# Exactly solvable Hamiltonian model of the doubled Ising and $\mathbb{Z}_2$ toric code topological phases separated by a gapped domain wall via anyon condensation

Yu Zhao[1, 2], Shan Huang[1, 2], Hongyu Wang[1, 2], Yuting Hu[3]$\star$ and Yidun Wan[1, 2]$\dagger$

**1** State Key Laboratory of Surface Physics, Department of Physics, Center for Field Theory and Particle Physics, and Institute for Nanoelectronic devices and Quantum computing, Fudan University, Shanghai 200433, China
**2** Shanghai Qi Zhi Institute, Shanghai 200030, China
**3** School of Physics, Hangzhou Normal University, Hangzhou 311121, China
$\star$ yuting.phys@gmail.com, $\dagger$ ydwan@fudan.edu.cn

February 8, 2023

## Abstract

In this paper, we construct an exactly solvable lattice Hamiltonian model of two topological phases separated by a gapped domain wall via anyon condensation. To be specific, we study the properties of this model in the case of the doubled Ising phase and $\mathbb{Z}_2$ toric code phase with a gapped domain wall in between. Our model is a concrete spatial counterpart of the phase transition triggered by anyon condensation, in the sense that the algebraically understood phenomena, such as splitting, identification, and confinement, in anyon condensation can be manifested in the spatial wavefunctions of our model. Our model also helps generalize the characteristic properties of a single topological phase: We find that the ground-state degeneracy of two topological phases separated by two gapped domain walls on the torus is equal to the number of quasiparticle types in the gapped domain walls; we also find the $S$ and $T$ matrices of our model.

# 1  Introduction

Relations between topological phases can be established via phase transitions [1–11] and via domain walls [9,12–22]. An interesting type of phase transitions between a topological

phase (the parent phase) and another (possibly trivial) topological phase (the child phase) is triggered by certain anyon condensation in the parent phase. As has been extensively studied [1, 6, 8], interesting phenomena occur during such a phase transition: 1) Certain anyons, including the condensed ones, may split into a few sectors, so more precisely speaking, a condensed anyon may not fully condense but only one or more sectors in its splitting condense. This is analogous to the Higgs boson condensation in breaking the electroweak symmetry, where only a one-dimensional subspace of the two-dimensional space the Higgs boson lives in condenses. 2) Since the condensed sectors become the new vacuum, two types of sectors related by fusing with a condensed anyon in the parent phase can no longer be distinguished in the child phase and thus are identified as the same type of anyons. 3) The anyons that have trivial (nontrivial) braiding statistics with the condensed anyons are unconfined (confined) in the child phase. 4) The child phase is a symmetry-enriched topological phase, with a hidden global symmetry. The hidden symmetry has been made precise in Ref. [11], which also proves a generalized Goldstone theorem of anyon condensation. Therefore, the phase transition from a parent topological phase to its child phase also belongs to the Landau-Ginzburg paradigm, however in a more general sense [8, 9, 11, 23].

Such a phase transition is temporal and is believed to correspond to a spatial gapped domain wall between the parent and child phases [9, 18]. Such domain walls have been studied algebraically via category theories [13, 16]. Since the Levin-Wen (LW) model [24], more specifically its extended version [25, 26] [1], is the most general model of 2-dimensional topological orders, can we possibly build a model that describes a parent phase and child phase with a gapped domain wall in between, based on the extended LW model? Such a Hamiltonian model would make it easy to study the properties of the gapped domain wall, the combined system, and the counterparts of the aforementioned phenomena in anyon condensation in the domain-wall picture dynamically, in terms of concrete wavefunctions.

On the other hand, it is profound to see whether the properties of a single topological phase would retain or how they would adapt to a system with two topological phases separated by a gapped domain wall. For example, the number of anyon species in a 2-dimensional topological phase coincides with the ground-state degeneracy of the topological phase on the torus [24, 27, 28]; the $S$ and $T$ matrices of the topological phase relates the basis transformations in the ground-state space to the mutual and self statistics of the anyon species [29–36]. Can we see similar properties in the case of a gapped domain wall? [2] The question can be answered if we have an explicit lattice Hamiltonian model of two topological phases separated by a gapped domain wall.

In this paper, we shall begin with an extended LW model describing a parent phase, and trigger the anyon condensation in half of the system to construct an exactly solvable lattice model describing the parent phase and its child phase separated by a gapped domain wall. We can certainly do this in a generic extended LW model. Nevertheless, to make various properties of the model specific and explicit, we shall only write down the model in the case of the doubled Ising and $\mathbb{Z}_2$ toric code topological phases with a gapped domain wall in between, since our discussion is easy to be generalized to general cases.

Our model has the same input data of the extended LW model that describes the parent phase, without using any extra categorical data, and is thus as simple as the original extended LW model.

---

[1]For our purpose to formulate the anyon condensation, we need the full spectrum of the elementary excitation states including the charge excitations that can not be defined in the original LW model but only in the extended version.

[2]A special case — topological phases with gapped boundaries — with these questions has been studied before [15, 37].

## 2   A brief review of the extended Levin-Wen model

Since our model is based on the extended LW model, we first briefly review the extended LW model. To be specific and for our purposes, we only consider the model that describes the doubled Ising phase.

The extended LW model is defined on a 2-dimensional honeycomb lattice (see Fig. 1a). Associated with each vertex is a tail, presented as a dangling edge near the vertex. It is arbitrary to choose the edge incident at the vertex to which to attach the tail because all choices are equivalent up to gauge transformations (see Appendix A).

The input data of the extended LW model describing the doubled Ising phase is a set $L_{\mathrm{DI}} = \{1, \sigma, \psi\}$, equipped with three functions $N : L_{\mathrm{DI}}^3 \to \mathbb{N}$, $d : L_{\mathrm{DI}} \to \mathbb{R}$, and $G : L_{\mathrm{DI}}^6 \to \mathbb{C}$. Each edge and tail of the lattice is labeled by an element in $L_{\mathrm{DI}}$.

The function $N$ sets the fusion rule and satisfies $N_{ij}^k = N_{ji}^k = N_{ik}^j$, whose nonzero independent elements are

$$N_{11}^1 = N_{\psi\psi}^1 = N_{\sigma\sigma}^1 = N_{\sigma\sigma}^\psi = 1. \tag{1}$$

The Hilbert space $\mathcal{H}$ is spanned by all possible assignments of the labels on the edges and tails, subject to the constraint $N_{ij}^k \neq 0$ on any three incident edges (tails) labeled by $i, j, k \in L_{\mathrm{DI}}$.

The function $d$ returns the quantum dimensions of the elements in $L_{\mathrm{DI}}$,

$$d_1 = d_\psi = 1, \qquad d_\sigma = \sqrt{2}. \tag{2}$$

The function $G$ has the symmetry $G_{kln}^{ijm} = G_{nkl}^{mij} = G_{ijn}^{klm}$. The nonzero independent elements are

$$G_{111}^{111} = G_{\psi\psi\psi}^{111} = G_{1\psi\psi}^{1\psi\psi} = 1, \quad G_{1\sigma\sigma}^{1\sigma\sigma} = G_{\psi\sigma\sigma}^{1\sigma\sigma} = -G_{\psi\sigma\sigma}^{\psi\sigma\sigma} = \frac{1}{\sqrt{2}}, \quad G_{\sigma\sigma\sigma}^{111} = G_{\sigma\sigma\sigma}^{1\psi\psi} = \frac{1}{\sqrt[4]{2}}. \tag{3}$$

The Hamiltonian of the extended LW model describing the doubled Ising phase reads

$$H_{\mathrm{DI}} := -\sum_V Q_V - \sum_P B_P^{\mathrm{DI}}. \tag{4}$$

The vertex operator $Q_V$ acts on the tail associated with vertex $V$ as

$$Q_V \left| \begin{array}{c} V \\ \vdash\!p \end{array} \right\rangle = \delta_{p,1} \left| \begin{array}{c} V \\ \vdash\!p \end{array} \right\rangle, \tag{5}$$

where $\delta$ denotes the Kronecker delta function that $\delta_{p,q} = 1$ if $p = q$ else $\delta_{p,q} = 0$.

The plaquette operator $B_P^{\mathrm{DI}}$ acting on plaquette $P$ is a sum:

$$B_P^{\mathrm{DI}} = \frac{d_1 B_P^1 + d_\sigma B_P^\sigma + d_\psi B_P^\psi}{4}, \tag{6}$$

where $B_P^s$, $s \in L_{\mathrm{DI}}$ is defined by

$$B_p^s \left| \begin{array}{c} \text{(hexagon diagram with labels } e_0, e_7, i_7, i_6, e_6, i_0, p, i_5, e_5, k_0, i_4, e_4, l_0, q, i_3, e_1, i_1, i_2, e_3, e_2) \end{array} \right\rangle = \delta_{p,1}\, \delta_{q,1} \sum_{j_0 j_1 j_2 j_3 j_4 j_5 j_6 j_7 \in L_{\mathrm{DI}}} \left( \prod_{n=0}^{7} \sqrt{d_{i_n} d_{j_n}} \right) \left( G_{sj_7j_0}^{e_0 i_0 i_7} \times \right.$$

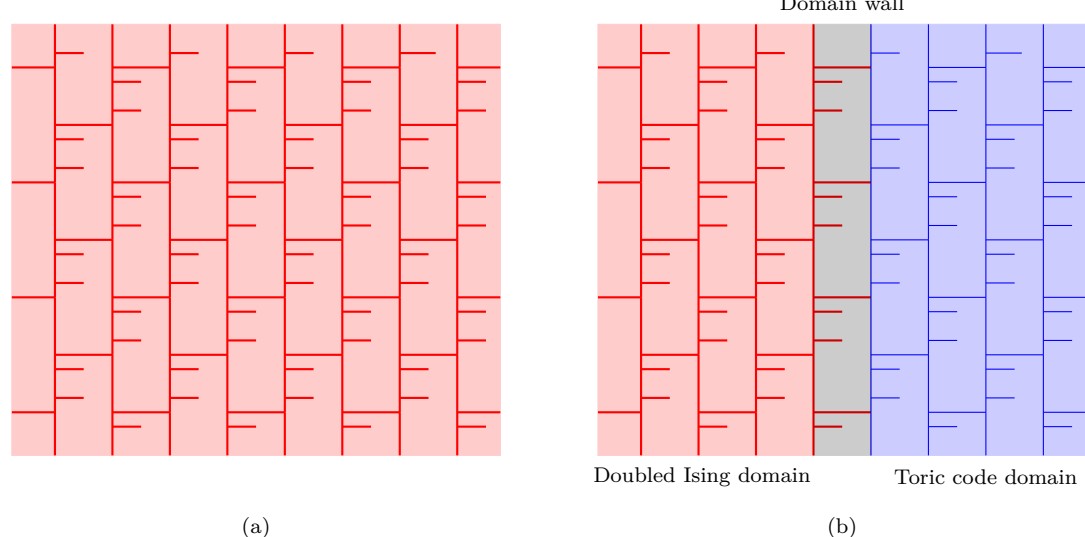

Figure 1: (a) The extended LW model describing the doubled Ising topological phase. (b) Our model of the doubled Ising and $\mathbb{Z}_2$ toric code topological phases separated by a gapped domain wall.

$$G^{e_1 i_1 i_0}_{s j_0 j_1}\ G^{e_2 i_2 i_1}_{s j_1 j_2}\ G^{e_3 i_3 i_2}_{s j_2 j_3}\ G^{e_4 i_4 i_3}_{s j_3 j_4}\ G^{e_5 i_5 i_4}_{s j_4 j_5}\ G^{e_6 i_6 i_5}_{s j_5 j_6}\ G^{e_7 i_7 i_6}_{s j_6 j_7}\Bigg) \left| \begin{array}{c} \text{diagram} \end{array} \right\rangle . \qquad (7)$$

The doubled-Ising Hamiltonian (4) is exactly solvable because all the summands $Q_V$ and $B_P^{\text{DI}}$ therein are commuting projectors.

## 3 The lattice model with a gapped domain wall between the doubled Ising and $\mathbb{Z}_2$ toric code phases

We now construct our model describing a doubled Ising phase and $\mathbb{Z}_2$ toric code phase separated by a gapped domain wall, via partial anyon condensation explained as follows. We divide the entire lattice into two halves, left and right. See Fig. 1b. Here, the edges and tails in the left (right) half are in red (blue); the plaquettes bounded by all red (blue) edges are in light red (blue); the plaquettes bounded by both red and blue edges are in gray. Each edge and tail of the entire lattice still take value in $L_{\text{DI}}$, so the Hilbert space of our model is still $\mathcal{H}$. We shall trigger anyon condensation in the right (blue) half, such that the doubled Ising phase therein will become the $\mathbb{Z}_2$ toric code phase through a phase transition.

Knowing that the $\mathbb{Z}_2$ toric code phase can be obtained by condensing $\psi\bar\psi$ anyons in the doubled Ising phase [8, 11, 12], we are motivated to add to the doubled-Ising Hamiltonian (4) the gapping term

$$\Delta H := -\Lambda \sum_{E\in\text{TC}} W_E^{\psi\bar\psi;1,1}, \qquad \Lambda \gg 1, \qquad (8)$$

where $E \in$ TC represents all the blue edges, and $W_E^{\psi\bar{\psi};1,1}$ is the creation operator of the $\psi\bar{\psi}$ anyons (to be defined in Section 4.1). The term $\Delta H$ renders the new ground states of the system the $+1$ eigenstates of the creation operators $W_E^{\psi\bar{\psi};1,1}\big|_{E \in \text{TC}}$, and thus are the superpositions of the states with arbitrarily many $\psi\bar{\psi}$ anyons in the right half of the lattice. We say that the $\psi\bar{\psi}$ anyons in the right half of the lattice are condensed. The total Hamiltonian now reads

$$H := -\sum_V Q_V - \sum_P B_P^{\text{DI}} + \Delta H. \tag{9}$$

By Eq. (88), the creation operators $W_E^{\psi\bar{\psi};1,1}$ of $\psi\bar{\psi}$ quasiparticle pairs can be written as

$$W_E^{\psi\bar{\psi};1,1} \left| \begin{matrix} i \\ j_E \\ k \end{matrix} \right\rangle = (-1)^{\delta_{j_E,\sigma}} \left| \begin{matrix} i \\ j_E \\ k \end{matrix} \right\rangle, \tag{10}$$

where $\delta$ is the Kronecker delta function. Hence, any blue edge has to overcome a great energy barrier ($\propto \Lambda$) to take value $\sigma$. For $\Lambda \to \infty$, the blue edges in the right half of the lattice effectively take value only in the input data $L_{\text{TC}} = \{1, \psi\} \subset L_{\text{DI}}$, equipped with the same $\delta$, $d$, and $G$ functions as that of $L_{\text{DI}}$ but restricted to $L_{\text{TC}}$:

$$\delta_{111} = \delta_{1\psi\psi} = 1, \qquad d_1 = d_\psi = 1, \qquad G_{111}^{111} = G_{\psi\psi\psi}^{111} = G_{1\psi\psi}^{1\psi\psi} = 1. \tag{11}$$

The right half of the system therefore describes the $\mathbb{Z}_2$ toric code phase [24, 26, 27], as a result of $\psi\bar{\psi}$ condensation.

Due to $\psi\bar{\psi}$ condensation, the effective Hilbert space $\mathcal{H}_{\text{eff}}$ of the model is the subspace of $\mathcal{H}$ in which all the blue edges can take value only in $L_{\text{TC}} = \{1, \psi\}$:

$$\mathcal{H}_{\text{eff}} := P_{\text{eff}} \mathcal{H}, \tag{12}$$

where $P_{\text{eff}}$ is the projector on the blue edges

$$P_{\text{eff}} := \prod_{E \in \text{TC}} (1 - \delta_{j_E,\sigma}) = \prod_{E \in \text{TC}} \frac{I + W_E^{\psi\bar{\psi};1,1}}{2}. \tag{13}$$

Hereafter, we refer to the red (blue) edges/plaquettes the DI (TC) edges/plaquettes. The gray plaquettes, bounded by both DI and TC edges, turn out to comprise the domain wall between the doubled Ising phase and $\mathbb{Z}_2$ toric code phase, and are thus called the DW plaquettes. Since $P_{\text{eff}} B_P^\sigma P_{\text{eff}} = 0$ if $P$ is a TC/DW plaquette, the effective plaquette operators acting on the DW and TC plaquettes in $\mathcal{H}_{\text{eff}}$ become

$$B_P^{\text{DW}} := P_{\text{eff}} B_P^{\text{DI}} P_{\text{eff}} = \frac{B_P^1 + B_P^\psi}{4}, \tag{14}$$

$$B_P^{\text{TC}} := P_{\text{eff}} B_P^{\text{DI}} P_{\text{eff}} = \frac{B_P^1 + B_P^\psi}{4}. \tag{15}$$

The effective Hamiltonian is the projection of the doubled-Ising Hamiltonian (4):

$$H_{\text{eff}} := P_{\text{eff}} H_{\text{DI}} P_{\text{eff}} = -\sum_V Q_V - \sum_{P \in \text{DI}} B_P^{\text{DI}} - \sum_{P \in \text{DW}} B_P^{\text{DW}} - \sum_{P \in \text{TC}} B_P^{\text{TC}}, \tag{16}$$

which is exactly solvable. The model describes the doubled Ising phase on the left, the $\mathbb{Z}_2$ toric code phase on the right, and a gapped domain wall in between. We shall refer to $H_{\text{eff}}$ as the Hamiltonian of our model from now on.

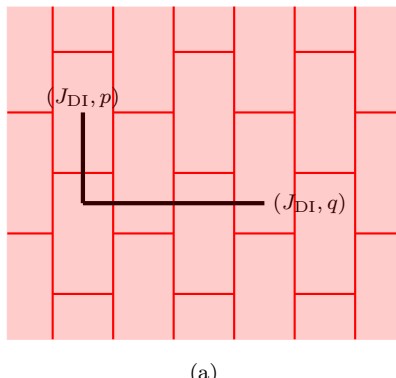 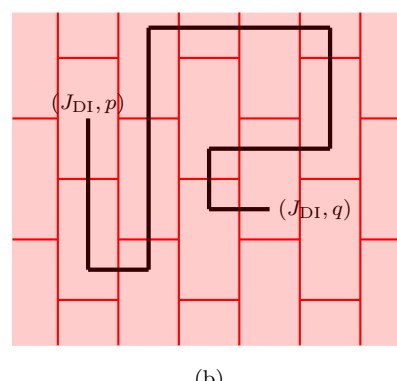

(a)

(b)

Figure 2: Two ribbon operators in the doubled Ising phase. The two ribbon operators both create quasiparticles $(J, p)$ and $(J, q)$ at the ends of their paths, which are homotopic. Hence, they are the same operators although they take very different paths.

## 4 The spectrum of the elementary excitation states

We now study the spectrum of our model. We assume the sphere topology, in which our model has a unique ground state $|\Phi\rangle$; nevertheless, the results in this section apply to other topologies. The ground state $|\Phi\rangle$ is defined by

$$Q_V |\Phi\rangle = B_P^{\mathrm{DI}} |\Phi\rangle = 2B_P^{\mathrm{TC}} |\Phi\rangle = 2B_P^{\mathrm{DW}} |\Phi\rangle = |\Phi\rangle \,, \tag{17}$$

where the factors 2 arise from the projection. In the ground state $|\Phi\rangle$, all the tails have to take value $1 \in L_{\mathrm{DI}}$.

An excited state $|\varphi\rangle$ is an eigenstate, in which $Q_V |\varphi\rangle = 0$ or $B_P^{\mathcal{D}} |\varphi\rangle = 0$ at one or more vertices $V$ or plaquettes $P$. In such a state, there are quasiparticles at vertices $V$ or plaquettes $P$. Here, the superscript $\mathcal{D}$ refers to either DW, TC, or DI. We also dub the ground state $|\Phi\rangle$ the trivial excited state, in which there are trivial quasiparticles.

For our purposes, it suffices to study the excited states with at most two quasiparticles, which we call *elementary excitation states*. Since in an elementary excitation state, all tails take value 1 except the ones where the two quasiparticles are, we can omit the tails irrelevant to these quasiparticles.

### 4.1 Review of the elementary excitation states in the doubled Ising phase

Since our model stems from the extended LW model describing the doubled Ising phase, we first focus on the elementary excitation states in the parent doubled Ising phase described by Hamiltonian $H_{\mathrm{DI}}$ (4). The doubled-Ising ground state $|\Phi\rangle_{\mathrm{DI}} \in \mathcal{H}$ satisfies

$$Q_V |\Phi\rangle_{\mathrm{DI}} = B_P^{\mathrm{DI}} |\Phi\rangle_{\mathrm{DI}} = |\Phi\rangle_{\mathrm{DI}} \tag{18}$$

for all vertices $V$ and plaquettes $P$. Each doubled-Ising elementary excitation state $|\varphi\rangle_{\mathrm{DI}}$ can be obtained by acting a ribbon operator $W_L$ on the ground state $|\Phi\rangle_{\mathrm{DI}}$ [24, 26]:

$$|\varphi\rangle_{\mathrm{DI}} = W_L |\Phi\rangle_{\mathrm{DI}} \,. \tag{19}$$

The ribbon operator $W_L$ is defined along a path $L$, which crosses one or more edges in the lattice, and creates a pair of quasiparticles at the two ends of $L$ (see Fig. 2). The path $L$ of a ribbon operator can be homotopically deformed, with its two ends fixed.

| Quasiparticle | Anyon species | Charge |
|:---:|:---:|:---:|
| $(1\bar{1}, 1)$ | $1\bar{1}$ | $1$ |
| $(1\bar{\sigma}, \sigma)$ | $1\bar{\sigma}$ | $\sigma$ |
| $(1\bar{\psi}, \psi)$ | $1\bar{\psi}$ | $\psi$ |
| $(\sigma\bar{1}, \sigma)$ | $\sigma\bar{1}$ | $\sigma$ |
| $(\sigma\bar{\sigma}, 1)$ | | $1$ |
| $(\sigma\bar{\sigma}, \psi)$ | $\sigma\bar{\sigma}$ | $\psi$ |
| $(\sigma\bar{\psi}, \sigma)$ | $\sigma\bar{\psi}$ | $\sigma$ |
| $(\psi\bar{1}, \psi)$ | $\psi\bar{1}$ | $\psi$ |
| $(\psi\bar{\sigma}, \sigma)$ | $\psi\bar{\sigma}$ | $\sigma$ |
| $(\psi\bar{\psi}, 1)$ | $\psi\bar{\psi}$ | $1$ |

Table 1: The anyon species and charges of quasiparticles in the doubled Ising phase.

We start with the elementary excitation states with a pair of quasiparticles in the two adjacent plaquettes with a common edge $E$. This state can be generated by ribbon operator $W_E^{J_{\text{DI}};p,q}$ along the shortest path that crosses only one edge $E$. This shortest ribbon operator creates in the two adjacent plaquettes a pair of quasiparticles $(J_{\text{DI}}, p)$ and $(J_{\text{DI}}, q)$, where $J_{\text{DI}}$ labels the anyon species of the quasiparticles, while $p$ and $q$ label the charges of the quasiparticles. Namely

$$|J_{\text{DI}}; p, q\rangle_{\text{DI}} = \left| \begin{array}{c} (J_{\text{DI}}, p) \rule{1cm}{0.4pt} (J_{\text{DI}}, q) \\ E \end{array} \right\rangle := W_E^{J_{\text{DI}};p,q} |\Phi\rangle_{\text{DI}}. \tag{20}$$

We only consider the action of ribbon operator $W_E^{J_{\text{DI}};p,q}$ on the ground state $|\Phi\rangle_{\text{DI}}$. The action reads

$$W_E^{J_{\text{DI}};p,q} \left| \begin{array}{c} j_E \end{array} \right\rangle = \sum_{k \in L_{\text{DI}}} \sqrt{\frac{d_k}{d_{j_E}}} \cdot \overline{z_{pqj_E}^{J_{\text{DI}};k}} \left| \begin{array}{c} j_E \\ p \underset{k}{\rule{0.6cm}{0.4pt}} q \\ j_E \end{array} \right\rangle, \tag{21}$$

where $j_E \in L_{\text{DI}}$ is the label on edge $E$. The matrix elements $z_{pqs}^{J_{\text{DI}};u}$ are the components of the tensor $z^{J_{\text{DI}}}$, and are listed in Appendix D.1. The tensor $z^{J_{\text{DI}}}$ satisfies [26]

$$\frac{\delta_{j,t} N_{rs}^t}{d_t} z_{pqt}^{J_{\text{DI}};w} = \sum_{ulv \in L_{\text{DI}}} d_u d_v z_{lqr}^{J_{\text{DI}};v} z_{pls}^{J_{\text{DI}};u} G_{pwu}^{rst} G_{qwv}^{srj} G_{rvw}^{sul}, \tag{22}$$

where the anyon species $J_{\text{DI}}$ labels different minimal solutions $z^{J_{\text{DI}}}$ that cannot be the sum of any other nonzero tensors. There are 9 anyon species:

$$1\bar{1}, \quad 1\bar{\sigma}, \quad 1\bar{\psi}, \quad \sigma\bar{1}, \quad \sigma\bar{\sigma}, \quad \sigma\bar{\psi}, \quad \psi\bar{1}, \quad \psi\bar{\sigma}, \quad \psi\bar{\psi}. \tag{23}$$

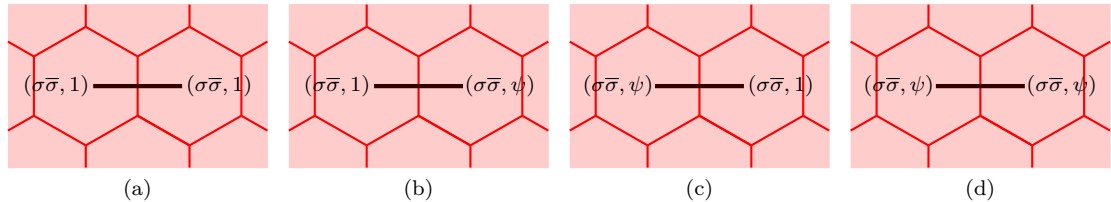

Figure 3: The doubled-Ising elementary excitation states with anyon species $\sigma\bar{\sigma}$. The charges of quasiparticles can take value arbitrarily in $\{1, \psi\}$.

Crossing any edge $E$ in the doubled Ising phase, for each $J_{\text{DI}} \neq \sigma\bar{\sigma}$, there is only one such ribbon operator $W_E^{J_{\text{DI}};p,q}$ with $p = q$. For $J_{\text{DI}} = \sigma\bar{\sigma}$, the corresponding charges $p$ and $q$ both can take values in $\{1, \psi\}$ (see Table 1); hence, there are four different such ribbon operators $W_E^{\sigma\bar{\sigma};p,q}$ with $p, q \in \{1, \psi\}$. All told, crossing any edge $E$, there are 12 shortest ribbon operators.

There are four degenerate elementary excitation states $|\sigma\bar{\sigma}; p, q\rangle_{\text{DI}} = W_E^{J_{\text{DI}};p,q} |\Phi\rangle_{\text{DI}}$ with $p, q \in \{1, \psi\}$, as shown in Fig. 3. Each state has two quasiparticles, each of which can be either $(\sigma\bar{\sigma}, 1)$ or $(\sigma\bar{\sigma}, \psi)$. While $\sigma\bar{\sigma}$ is the topological observable of these states, the degenerate charges 1 and $\psi$ cannot be distinguished experimentally [26], as they can be transformed into each other by local operators $B_P^{1\sigma\psi\sigma}$ and $B_P^{\psi\sigma1\sigma}$ (defined in Eq. (111)). The elementary excitation states in the doubled Ising phase hence are characterized by their anyon species.

Then we study the states with two quasiparticles in two nonadjacent plaquettes, generated by ribbon operators along longer paths. These ribbon operators result from concatenating shorter ribbon operators. For example, in Fig. 4, the two shortest ribbon operators create in plaquette $P$ two identical quasiparticles $(J_{\text{DI}}, q)$, which are then annihilated, resulting in a longer ribbon operator $W_L^{J_{\text{DI}};p,r}$, which generates an elementary excitation state $W_L^{J_{\text{DI}};p,r} |\Phi\rangle_{\text{DI}}$ with two quasiparticles at the end of path $L$. The matrix elements of such ribbon operators are also given by $z^{J_{\text{DI}}}$ tensors (see Appendix B.2).

### 4.2 The elementary excitation states of our model

Now we study the elementary excitation states of our model. These states are eigenstates of the effective Hamiltonian $H_{\text{eff}}$ (16) in the effective Hilbert space $\mathcal{H}_{\text{eff}}$ (12).

According to Appendix C, the projector (13), $P_{\text{eff}}$, commutes with any doubled-Ising ribbon operator $W_L^{J_{\text{DI}};p,q}$ in the effective Hilbert space $\mathcal{H}_{\text{eff}}$:

$$P_{\text{eff}} \left[ W_L^{J_{\text{DI}};p,q}, \ P_{\text{eff}} \right] = 0. \tag{24}$$

Then, together with Eq. (19), $P_{\text{eff}}$ projects the elementary excitation states $|\varphi\rangle_{\text{DI}}$ and the ribbon operators $W_L^{J_{\text{DI}};p,q}$ of the doubled Ising phase to those of our model. While in the doubled Ising phase, elementary excitations states do not discern the locations of the quasiparticles but only their anyon species, in our model, nevertheless, locations of the quasiparticles do matter because of the domain wall (see Fig. 5). In what follows, we shall study the elementary excitation states of our model in the cases of different quasiparticle locations.

#### 4.2.1 The elementary excitation states with quasiparticle pairs in the toric code domain

Here, we study the elementary excitation states of our model with a pair of quasiparticles in two adjacent plaquettes completely in the toric code domain. These states result from



Figure 4: Concatenating two shortest ribbon operators to a longer one. (a) The state generated by two shortest ribbon operators. (b) Annihilating the two quasiparticles $(J_{\mathrm{DI}}, q)$ in plaquette $P$ results in (c): the state generated by the longer ribbon operator $W_L^{J_{\mathrm{DI}};p,r}$.

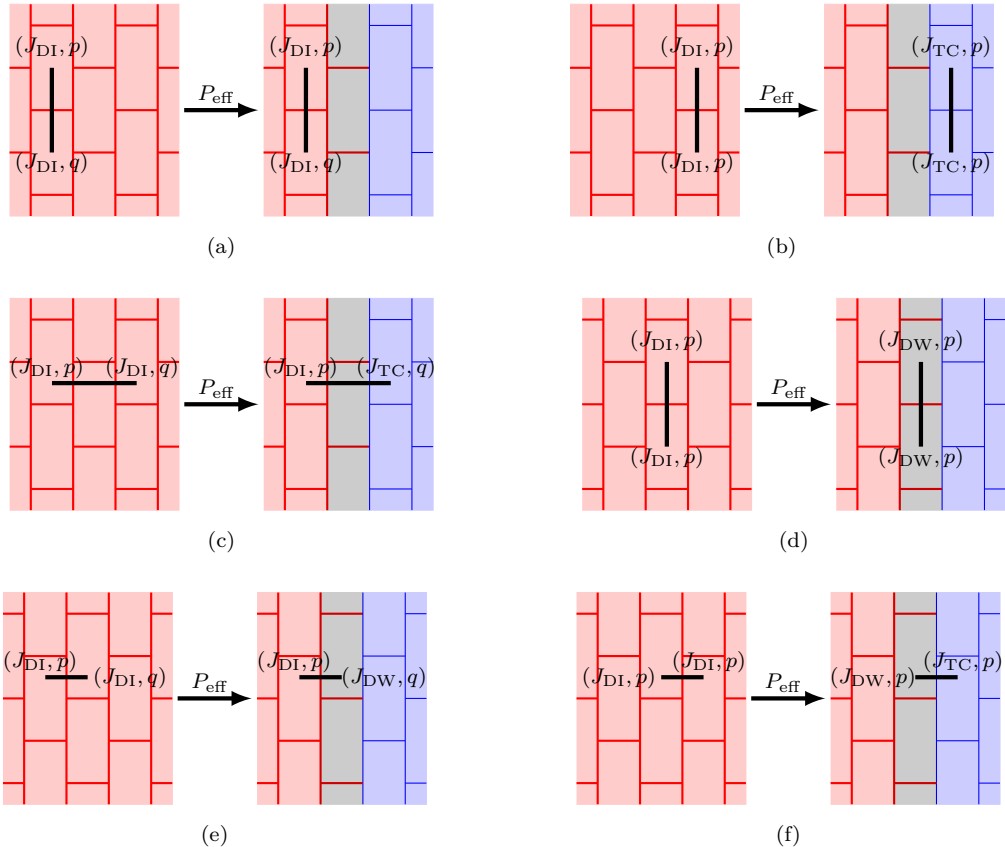

Figure 5: Projecting the doubled-Ising elementary excitation states results in the elementary excitation states in our model. (a) The two quasiparticles in our model are all in the doubled Ising domain. (b) The two quasiparticles are all in the toric code domain. (c) The two quasiparticles are respectively in the doubled Ising domain and the toric code domain. (d) The two quasiparticles are all in the gapped domain wall. (e) The two quasiparticles are respectively in the doubled Ising domain and the gapped domain wall. (f) The two quasiparticles are respectively in the toric code domain and the gapped domain wall.

| Quasiparticle | Anyon species | Charge |
|:---:|:---:|:---:|
| $(1,1)$ | $1$ | $1$ |
| $(e,\psi)$ | $e$ | $\psi$ |
| $(m,1)$ | $m$ | $1$ |
| $(\epsilon,\psi)$ | $\epsilon$ | $\psi$ |

Table 2: The anyon species and charges of quasiparticles in the toric code domain.

projecting the elementary excitation states $|J_{\mathrm{DI}};p,q\rangle_{\mathrm{DI}}$ (20) in the doubled Ising phase:

$$
\begin{aligned}
|1;1,1\rangle &:= P_{\mathrm{eff}}|1\bar{1};1,1\rangle_{\mathrm{DI}} = P_{\mathrm{eff}}\left|\psi\bar{\psi};1,1\right\rangle_{\mathrm{DI}}, \\
|\epsilon;\psi,\psi\rangle &:= P_{\mathrm{eff}}|\psi\bar{1};\psi,\psi\rangle_{\mathrm{DI}} = P_{\mathrm{eff}}\left|1\bar{\psi};\psi,\psi\right\rangle_{\mathrm{DI}}, \\
|m;1,1\rangle &:= P_{\mathrm{eff}}|\sigma\bar{\sigma};1,1\rangle_{\mathrm{DI}}, \\
|e;\psi,\psi\rangle &:= P_{\mathrm{eff}}|\sigma\bar{\sigma};\psi,\psi\rangle_{\mathrm{DI}},
\end{aligned}
\tag{25}
$$

where we define the four nonvanishing states after the projection as $|J_{\mathrm{TC}};p,p\rangle$, with $J_{\mathrm{TC}} \in \{1,\epsilon,e,m\}$ the anyon species and $p \in \{1,\psi\}$ the charges of the quasiparticles. See Fig. 5b. But not all doubled-Ising elementary excitation states are projected to states in $\mathcal{H}_{\mathrm{eff}}$:

$$
\begin{aligned}
P_{\mathrm{eff}}|\sigma\bar{\sigma};\psi,1\rangle_{\mathrm{DI}} &= P_{\mathrm{eff}}|\sigma\bar{\sigma};1,\psi\rangle_{\mathrm{DI}} = 0, \\
P_{\mathrm{eff}}|\sigma\bar{1};\sigma,\sigma\rangle_{\mathrm{DI}} &= 0, \\
P_{\mathrm{eff}}\left|\sigma\bar{\psi};\sigma,\sigma\right\rangle_{\mathrm{DI}} &= 0, \\
P_{\mathrm{eff}}|1\bar{\sigma};\sigma,\sigma\rangle_{\mathrm{DI}} &= 0, \\
P_{\mathrm{eff}}|\psi\bar{\sigma};\sigma,\sigma\rangle_{\mathrm{DI}} &= 0.
\end{aligned}
\tag{26}
$$

These states are infinitely ($\Lambda \to \infty$) gapped by $\Delta H$, and should not appear in $\mathcal{H}_{\mathrm{eff}}$.

The four elementary excitation states $|J_{\mathrm{TC}};p,p\rangle$ are precisely the known four elementary excitation states in the $\mathbb{Z}_2$ toric code phase. These states are generated by the ribbon operators $W_E^{J_{\mathrm{TC}};p,p}$ acting on the ground state (17) $|\Phi\rangle$ of our model

$$
|J_{\mathrm{TC}};p,p\rangle = \left| \vphantom{\sum} \quad\right\rangle = W_E^{J_{\mathrm{TC}};p,p}|\Phi\rangle,
\tag{27}
$$

where the ribbon operators are the projections

$$
\begin{aligned}
W_E^{1;1,1} &:= P_{\mathrm{eff}}W_E^{1\bar{1};1,1}P_{\mathrm{eff}} = P_{\mathrm{eff}}W_E^{\psi\bar{\psi};1,1}P_{\mathrm{eff}}, \\
W_E^{\epsilon;\psi,\psi} &:= P_{\mathrm{eff}}W_E^{\psi\bar{1};\psi,\psi}P_{\mathrm{eff}} = P_{\mathrm{eff}}W_E^{1\bar{\psi};\psi,\psi}P_{\mathrm{eff}}, \\
W_E^{m;1,1} &:= P_{\mathrm{eff}}W_E^{\sigma\bar{\sigma};1,1}P_{\mathrm{eff}}, \\
W_E^{e;\psi,\psi} &:= P_{\mathrm{eff}}W_E^{\sigma\bar{\sigma};\psi,\psi}P_{\mathrm{eff}}.
\end{aligned}
\tag{28}
$$

Note that the two plaquettes sharing edge $E$ must lie within the toric code domain.

| Elementary excitation state | Doubled-Ising quasiparticle | Toric-code quasiparticle |
|:---:|:---:|:---:|
| $\lvert 1\bar{1}\text{-}1;1,1\rangle$ | $(1\bar{1},1)$ | |
| $\lvert \psi\bar{\psi}\text{-}1;1,1\rangle$ | $(\psi\bar{\psi},1)$ | $(1,1)$ |
| $\lvert \psi\bar{1}\text{-}\epsilon;\psi,\psi\rangle$ | $(\psi\bar{1},\psi)$ | |
| $\lvert 1\bar{\psi}\text{-}\epsilon;\psi,\psi\rangle$ | $(1\bar{\psi},\psi)$ | $(\epsilon,\psi)$ |
| $\lvert \sigma\bar{\sigma}\text{-}m;1,1\rangle$ | $(\sigma\bar{\sigma},1)$ | |
| $\lvert \sigma\bar{\sigma}\text{-}m;\psi,1\rangle$ | $(\sigma\bar{\sigma},\psi)$ | $(m,1)$ |
| $\lvert \sigma\bar{\sigma}\text{-}e;1,\psi\rangle$ | $(\sigma\bar{\sigma},1)$ | |
| $\lvert \sigma\bar{\sigma}\text{-}e;\psi,\psi\rangle$ | $(\sigma\bar{\sigma},\psi)$ | $(e,\psi)$ |

Table 3: The interdomain elementary excitation states.

These ribbon operators $W_E^{J_{\mathrm{TC}};p,q}$ read

$$
W_E^{J_{\mathrm{TC}};p,q}\left\lvert \begin{array}{c} j_E \end{array} \right\rangle = \sum_{k\in L_{\mathrm{TC}}} \sqrt{\frac{d_k}{d_{j_E}}}\cdot \overline{z_{ppj_E}^{J_{\mathrm{TC}};k}}\left\lvert \begin{array}{c} j_E \ \ p \\ p\ \ k \ \ j_E \end{array} \right\rangle. \tag{29}
$$

The components $z_{ppj_E}^{J_{\mathrm{TC}};k}$ are listed in Appendix D.2 and are precisely those comprising the ribbon operators in the extended LW model describing the $\mathbb{Z}_2$ toric code phase [26].

Ribbon operators defined along longer paths crossing edges in the toric code domain can be obtained also by concatenating shorter ribbon operators. We shall not dwell on this.

### 4.2.2 The elementary excitation states with interdomain quasiparticle pairs

Now we consider what we call *interdomain elementary excitation states*, each having one quasiparticle $(J_{\mathrm{DI}},p)$ in the doubled Ising domain and the other $(J_{\mathrm{TC}},q)$ in the toric code domain. An interdomain state bears two different topological observables, $J_{\mathrm{DI}}$ in the doubled Ising domain and $J_{\mathrm{TC}}$ in the toric code domain. See Fig. 5c. We label interdomain elementary excitation states as $\lvert J_{\mathrm{DI}}\text{-}J_{\mathrm{TC}};p,q\rangle$.

There are 8 distinct interdomain elementary excitation states, as listed in Table 3.

These states can be generated by the ribbon operators along paths $L$ across the gapped domain wall:

$$
\lvert J_{\mathrm{DI}}\text{-}J_{\mathrm{TC}};p,q\rangle = \left\lvert \begin{array}{c} (J_{\mathrm{DI}},p)\ \ \ \ \ \underset{E_1}{\overset{L\ \ E_4}{\rule{2cm}{0.4pt}}}\ \ \ \ \ (J_{\mathrm{TC}},q) \end{array} \right\rangle
$$

$$
= W_L^{J_{\mathrm{DI}}\text{-}J_{\mathrm{TC}};p,q}\left\lvert \begin{array}{c} E_1 \ \ \ \ \ \ \ E_4 \end{array} \right\rangle. \tag{30}
$$

| Quasiparticle | Anyon species | Charge |
|:---:|:---:|:---:|
| $(1,1)$ | $1$ | $1$ |
| $(e,\psi)$ | $e$ | $\psi$ |
| $(m,1)$ | $m$ | $1$ |
| $(\epsilon,\psi)$ | $\epsilon$ | $\psi$ |
| $(\chi,\sigma)$ | $\chi$ | $\sigma$ |
| $(\overline{\chi},\sigma)$ | $\overline{\chi}$ | $\sigma$ |

Table 4: The quasiparticle species and charges of the quasiparticles in the gapped domain wall.

The ribbon operators $W_L^{J_{\mathrm{DI}} \text{-} J_{\mathrm{TC}};p,q}$ are projected from the doubled-Ising ribbon operators along same paths $L$:

$$W_L^{J_{\mathrm{DI}} \text{-} J_{\mathrm{TC}};p,q} := P_{\mathrm{eff}} W_L^{J_{\mathrm{DI}};p,q} P_{\mathrm{eff}}, \tag{31}$$

and are explicitly written as

$$\tag{32}$$

### 4.2.3   The elementary excitation states with domainwall quasiparticle pairs

Now we study the *domainwall elementary excitation states*, i.e., the elementary excitation states with a pair of quasiparticles in two adjacent plaquettes within the gapped domain wall (see Fig. 5d). These states are projected from the doubled-Ising elementary excitation states with a pair of quasiparticles in the same plaquettes. Namely,

$$
\begin{aligned}
|1;1,1\rangle &:= P_{\mathrm{eff}} |1\bar{1};1,1\rangle_{\mathrm{DI}} = P_{\mathrm{eff}} \left|\psi\bar{\psi};1,1\right\rangle_{\mathrm{DI}}, \\
|\epsilon;\psi,\psi\rangle &:= P_{\mathrm{eff}} |\psi\bar{1};\psi,\psi\rangle_{\mathrm{DI}} = P_{\mathrm{eff}} \left|1\bar{\psi};\psi,\psi\right\rangle_{\mathrm{DI}}, \\
|\chi;\sigma,\sigma\rangle &:= P_{\mathrm{eff}} |\sigma\bar{1};\sigma,\sigma\rangle_{\mathrm{DI}} = P_{\mathrm{eff}} \left|\sigma\bar{\psi};\sigma,\sigma\right\rangle_{\mathrm{DI}}, \\
|\bar{\chi};\sigma,\sigma\rangle &:= P_{\mathrm{eff}} |1\bar{\sigma};\sigma,\sigma\rangle_{\mathrm{DI}} = P_{\mathrm{eff}} |\psi\bar{\sigma};\sigma,\sigma\rangle_{\mathrm{DI}}, \\
|m;1,1\rangle &:= P_{\mathrm{eff}} |\sigma\bar{\sigma};1,1\rangle_{\mathrm{DI}}, \\
|e;\psi,\psi\rangle &:= P_{\mathrm{eff}} |\sigma\bar{\sigma};\psi,\psi\rangle_{\mathrm{DI}},
\end{aligned}
\tag{33}
$$

where we define the six nonvanishing states after the projection as $|J_{\mathrm{DW}};p,p\rangle$, with $J_{\mathrm{DW}} \in \{1,\epsilon,m,e,\chi,\bar{\chi}\}$ the quasiparticle species and $p \in L_{\mathrm{DI}}$ the charges of the quasi-

particles. Graphically,

$$|J_{\text{DW}}; p, p\rangle = \left| \begin{array}{c} (J_{\text{DW}}, p) \rule[0.5ex]{2em}{1.5pt} (J_{\text{DW}}, p) \end{array} \right\rangle . \qquad (34)$$

Although there are 6 distinct domain wall elementary excitation states (33), there are 10 different ribbon operators across an edge $E$ in the gapped domain wall:

$$W_{E,1}^{1;1,1} := P_{\text{eff}} W_E^{1\bar{1};1,1} P_{\text{eff}}, \qquad W_{E,2}^{1;1,1} := P_{\text{eff}} W_E^{\psi\bar{\psi};1,1} P_{\text{eff}},$$

$$W_{E,1}^{\epsilon;\psi,\psi} := P_{\text{eff}} W_E^{\psi\bar{1};\psi,\psi} P_{\text{eff}}, \qquad W_{E,2}^{\epsilon;\psi,\psi} := P_{\text{eff}} W_E^{1\bar{\psi};\psi,\psi} P_{\text{eff}},$$

$$W_{E,1}^{\chi;\sigma,\sigma} := P_{\text{eff}} W_E^{\sigma\bar{1};\sigma,\sigma} P_{\text{eff}}, \qquad W_{E,2}^{\chi;\sigma,\sigma} := P_{\text{eff}} W_E^{\sigma\bar{\psi};\sigma,\sigma} P_{\text{eff}},$$

$$W_{E,1}^{\bar{\chi};\sigma,\sigma} := P_{\text{eff}} W_E^{1\bar{\sigma};\sigma,\sigma} P_{\text{eff}}, \qquad W_{E,2}^{\bar{\chi};\sigma,\sigma} := P_{\text{eff}} W_E^{\psi\bar{\sigma};\sigma,\sigma} P_{\text{eff}},$$

$$W_E^{m;1,1} := P_{\text{eff}} W_E^{\sigma\bar{\sigma};1,1} P_{\text{eff}},$$

$$W_E^{e;\psi,\psi} := P_{\text{eff}} W_E^{\sigma\bar{\sigma};\psi,\psi} P_{\text{eff}}. \qquad (35)$$

Since $E$ is a DI edge taking value in $L_{\text{DI}} = \{1, \psi, \sigma\}$,

$$W_{E,1}^{J_{\text{DW}}; p, p} \neq W_{E,2}^{J_{\text{DW}}; p, p} \qquad (36)$$

for $J_{\text{DW}} = 1, \epsilon, \chi$ and $\bar{\chi}$, but

$$W_{E,1}^{J_{\text{DW}}; p, p} |\Phi\rangle = W_{E,2}^{J_{\text{DW}}; p, p} |\Phi\rangle = |J_{\text{DW}}; p, p\rangle . \qquad (37)$$

Specifically,

$$W_{E,i}^{J_{\text{DW}}; p, p} \left| \begin{array}{c} j_E \end{array} \right\rangle = \sum_{k \in L_{\text{DI}}} \sqrt{\frac{d_k}{d_{j_E}}} \cdot \overline{z_{ppj_E}^{J_{\text{DW}};k}} \left| \begin{array}{c} j_E \\ p \rule{1em}{0.4pt} k \rule{1em}{0.4pt} p \\ j_E \end{array} \right\rangle , \qquad (38)$$

where the coefficients $z_{pqj_E}^{J_{\text{DW}};k}$ are listed in Appendix D.3. The indices $p, q, k \in L_{\text{DI}}$, but the index $j_E$ is restricted to $L_{\text{TC}} = \{1, \psi\}$ because edge $E$ only takes value in $L_{\text{TC}}$ in the ground state (17) $|\Phi\rangle$.

### 4.2.4 The elementary excitation states with doubled-Ising-domainwall quasiparticle pairs

We now consider the elementary excitation states with one doubled-Ising quasiparticle $(J_{\text{DI}}, p)$ and one domainwall quasiparticle $(J_{\text{DW}}, q)$ in the adjacent plaquettes. See Fig. 5e. These elementary excitation states are defined as

$$|J_{\text{DI}}\text{-}J_{\text{DW}}; p, q\rangle := W_E^{J_{\text{DI}}; p, q} |\Phi\rangle , \qquad (39)$$

where $W_E^{J_{\text{DI}}; p, q}$ is the ribbon operator across a DI edge $E$ between the doubled Ising domain and the gapped domain wall.

$$W_E^{J_{\text{DI}}; p, q} \left| \begin{array}{c} j_E \end{array} \right\rangle = \sum_{k \in L_{\text{DI}}} \sqrt{\frac{d_k}{d_{j_E}}} \cdot \overline{z_{pqj_E}^{J_{\text{DI}};k}} \left| \begin{array}{c} j_E \\ p \rule{1em}{0.4pt} k \rule{1em}{0.4pt} q \\ j_E \end{array} \right\rangle . \qquad (40)$$

There are 12 possible distinct elementary excitation states, as in Table 5.

| Elementary excitation state | Doubled-Ising quasiparticle | Domainwall quasiparticle |
|---|---|---|
| $\left|1\bar{1}\text{-}1;1,1\right\rangle$ | $(1\bar{1},1)$ | $(1,1)$ |
| $\left|\psi\bar{\psi}\text{-}1;1,1\right\rangle$ | $(\psi\bar{\psi},1)$ | |
| $\left|\psi\bar{1}\text{-}\epsilon;\psi,\psi\right\rangle$ | $(\psi\bar{1},\psi)$ | $(\epsilon,\psi)$ |
| $\left|1\bar{\psi}\text{-}\epsilon;\psi,\psi\right\rangle$ | $(1\bar{\psi},\psi)$ | |
| $\left|\sigma\bar{1}\text{-}\chi;\sigma,\sigma\right\rangle$ | $(\sigma\bar{1},\sigma)$ | $(\chi,\sigma)$ |
| $\left|\sigma\bar{\psi}\text{-}\chi;\sigma,\sigma\right\rangle$ | $(\sigma\bar{\psi},\sigma)$ | |
| $\left|1\bar{\sigma}\text{-}\bar{\chi};\sigma,\sigma\right\rangle$ | $(1\bar{\sigma},\sigma)$ | $(\bar{\chi},\sigma)$ |
| $\left|\psi\bar{\sigma}\text{-}\bar{\chi};\sigma,\sigma\right\rangle$ | $(\psi\bar{\sigma},\sigma)$ | |
| $\left|\sigma\bar{\sigma}\text{-}m;1,1\right\rangle$ | $(\sigma\bar{\sigma},1)$ | $(m,1)$ |
| $\left|\sigma\bar{\sigma}\text{-}m;\psi,1\right\rangle$ | $(\sigma\bar{\sigma},\psi)$ | |
| $\left|\sigma\bar{\sigma}\text{-}e;1,\psi\right\rangle$ | $(\sigma\bar{\sigma},1)$ | $(e,\psi)$ |
| $\left|\sigma\bar{\sigma}\text{-}e;\psi,\psi\right\rangle$ | $(\sigma\bar{\sigma},\psi)$ | |

Table 5: The elementary excitation states with one doubled-Ising quasiparticle and one domainwall quasiparticle.

### 4.2.5 The elementary excitation states with toric-code-domainwall quasiparticle pairs

We consider the case in Fig. 5f: the elementary excitation states with one toric-code quasiparticle and one domainwall quasiparticle. These elementary excitation states are defined as

$$|J_{\mathrm{DW}}\text{-}J_{\mathrm{TC}};p,q\rangle := W_E^{J_{\mathrm{TC}};p,p}|\Phi\rangle, \tag{41}$$

where $W_E^{J_{\mathrm{TC}};p,p}$ is the ribbon operator across TC edge $E$ between the toric code domain and the gapped domain wall.

$$W_E^{J_{\mathrm{TC}};p,p}\left|\,\includegraphics{}\,\right\rangle = \sum_{k \in L_{\mathrm{TC}}}\sqrt{\frac{d_k}{d_{j_E}}} \cdot \overline{z_{ppj_E}^{J_{\mathrm{TC}};k}}\left|\,\includegraphics{}\,\right\rangle. \tag{42}$$

There are 4 distinct elementary excitation states generated by $W_E^{J_{\mathrm{TC}};p,q}$, as in Table 6.

Concatenating the ribbon operators in (40) and (42) results in the interdomain ribbon operators (31).

## 5 Correspondence with anyon condensation

As mentioned in the introduction, our model of two topological phases separated by a gapped domain wall can be regarded as a spatial counterpart of the phase transition (which is temporal) from one phase (the parent phase) to the other (the child phase) triggered by anyon condensation. See Fig. 6.

An intermediate phase during the phase transition was introduced as merely a method to study the procedure of anyon condensation [1]. The anyon condensation in a parent

| Elementary excitation state | Domainwall quasiparticle | Toric-code quasiparticle |
|:---:|:---:|:---:|
| $\lvert 1\text{-}1; 1, 1\rangle$ | $(1, 1)$ | $(1, 1)$ |
| $\lvert e\text{-}e; \psi, \psi\rangle$ | $(e, \psi)$ | $(e, \psi)$ |
| $\lvert m\text{-}m; 1, 1\rangle$ | $(m, 1)$ | $(m, 1)$ |
| $\lvert \epsilon\text{-}\epsilon; \psi, \psi\rangle$ | $(\epsilon, \psi)$ | $(\epsilon, \psi)$ |

Table 6: The elementary excitation states with one toric-code quasiparticle and one domainwall quasiparticle.

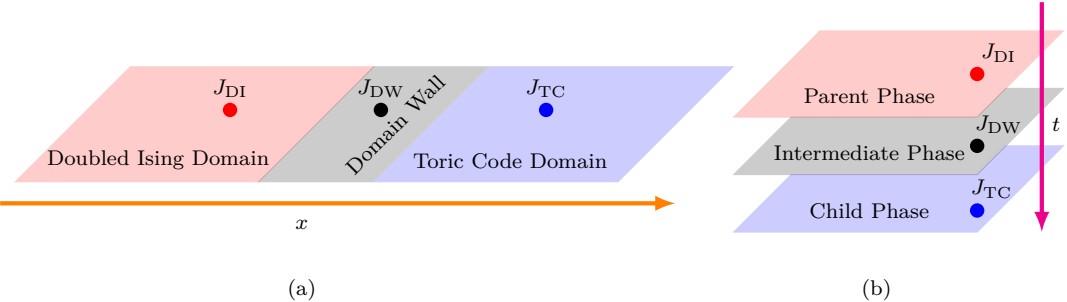

(a)                                              (b)

Figure 6: The correspondence between (a): our model with gapped domain wall between the doubled Ising domain and toric code domain and (b): the anyon condensation from the parent doubled Ising phase to the child toric code phase via an auxiliary intermediate phase.

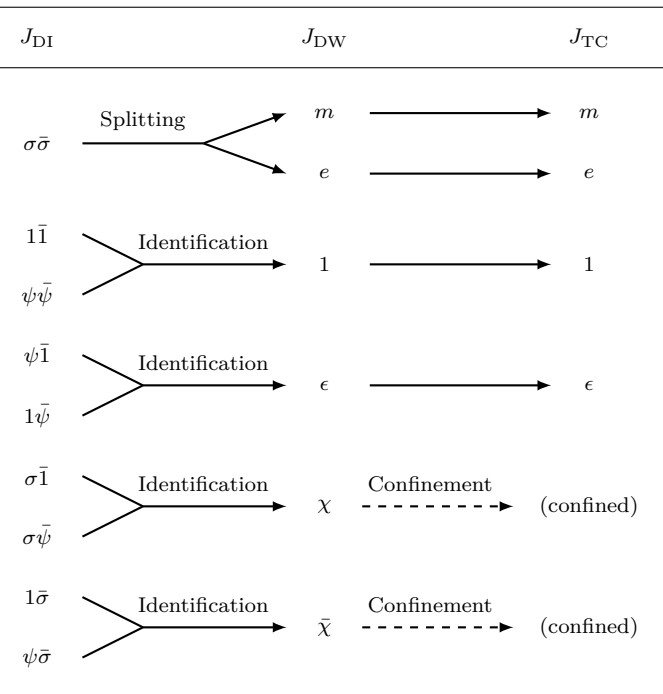

Figure 7: Relations between the quasiparticle species in different domains (phases).

| Phenomenon | Projection | Interdomain states |
|---|---|---|
| Splitting | $P_{\text{eff}} \lvert \sigma\bar{\sigma}; 1, 1\rangle_{\text{DI}} = \lvert m; 1, 1\rangle$ | $\lvert \sigma\bar{\sigma}\text{-}m; 1, 1\rangle, \quad \lvert \sigma\bar{\sigma}\text{-}m; \psi, 1\rangle$ |
| | $P_{\text{eff}} \lvert \sigma\bar{\sigma}; \psi, \psi\rangle_{\text{DI}} = \lvert e; \psi, \psi\rangle$ | $\lvert \sigma\bar{\sigma}\text{-}e; 1, \psi\rangle, \quad \lvert \sigma\bar{\sigma}\text{-}e; \psi, \psi\rangle$ |
| Identification | $P_{\text{eff}} \lvert 1\bar{1}; 1, 1\rangle_{\text{DI}} = \lvert 1; 1, 1\rangle$ | $\lvert 1\bar{1}\text{-}1; 1, 1\rangle$ |
| | $P_{\text{eff}} \lvert \psi\bar{\psi}; 1, 1\rangle_{\text{DI}} = \lvert 1; 1, 1\rangle$ | $\lvert \psi\bar{\psi}\text{-}1; 1, 1\rangle$ |
| | $P_{\text{eff}} \lvert \psi\bar{1}; \psi, \psi\rangle_{\text{DI}} = \lvert \epsilon; \psi, \psi\rangle$ | $\lvert \psi\bar{1}\text{-}\epsilon; \psi, \psi\rangle$ |
| | $P_{\text{eff}} \lvert 1\bar{\psi}; \psi, \psi\rangle_{\text{DI}} = \lvert \epsilon; \psi, \psi\rangle$ | $\lvert 1\bar{\psi}\text{-}\epsilon; \psi, \psi\rangle$ |
| Confinement | $P_{\text{eff}} \lvert \sigma\bar{1}; \sigma, \sigma\rangle_{\text{DI}} = 0$ | |
| | $P_{\text{eff}} \lvert \sigma\bar{\psi}; 1, 1\rangle_{\text{DI}} = 0$ | There is no interdomain state |
| | $P_{\text{eff}} \lvert 1\bar{\sigma}; \sigma, \sigma\rangle_{\text{DI}} = 0$ | with $J_{\text{DI}} = \sigma\bar{1}, \ \sigma\bar{\psi}, \ 1\bar{\sigma}, \ \psi\bar{\sigma}$. |
| | $P_{\text{eff}} \lvert \psi\bar{\sigma}; \sigma, \sigma\rangle_{\text{DI}} = 0$ | |

Table 7: The projection-state correspondence in the phenomena of splitting, identification, and confinement in the toric code domain.

phase first leads to an intermediate phase where splitting and identification have been completed, while the confinement occurs during the transition from the intermediate phase to the child phase. Interestingly, this auxiliary, virtual intermediate phase corresponds to the physical gapped domain wall between the parent and child phases. For example, Figure 7 records the relations between the quasiparticles in different domains in our model, corresponding to different stages in a phase transition induced by $\psi\bar{\psi}$ condensation in the doubled Ising phase. Here, we shall use our model to formulate this correspondence rigorously.

The three main phenomena — splitting, identification, and confinement — that occur in a phase transition due to anyon condensation can find their spatial counterparts in the elementary excitation states of our model. In Eq. (25), the doubled-Ising elementary excitation states $\lvert J_{\text{DI}}; p, p\rangle_{\text{DI}}$ in the parent phase are projected to the states $\lvert J_{\text{TC}}; p, p\rangle$ with quasiparticles in the toric code domain of our model. On the other hand, any interdomain elementary excitation state (30) $\lvert J_{\text{DI}}\text{-}J_{\text{TC}}; p, q\rangle$ bears a pair of topological observables $J_{\text{DI}}\text{-}J_{\text{TC}}$. The allowed pairs $J_{\text{DI}}\text{-}J_{\text{TC}}$ in the interdomain elementary excitation states are in one-to-one correspondence with the projections from $\lvert J_{\text{DI}}; p, q\rangle_{\text{DI}}$ to $\lvert J_{\text{TC}}; p, q\rangle$. Table 7 records this correspondence. We dub this correspondence the *projection-state correspondence*. We now exhibit this correspondence from three aspects: splitting, identification, and confinement.

## 5.1 Splitting

Seen in Table 7, the originally indistinguishable elementary excitation states $\lvert \sigma\bar{\sigma}; 1, 1\rangle_{\text{DI}}$ and $\lvert \sigma\bar{\sigma}; \psi, \psi\rangle_{\text{DI}}$ are projected to the topological different states $\lvert m; 1, 1\rangle$ and $\lvert e; \psi, \psi\rangle$ via $\psi\bar{\psi}$ condensation. It appears that the anyon species $\sigma\bar{\sigma}$ in the doubled Ising phase 'splits' into two anyon species $e$ and $m$ in the toric code domain. This phenomenon is precisely what is known as *splitting* in the language of anyon condensation.

The phenomenon of splitting can also be seen spatially in the interdomain elementary excitation states under the projection-state correspondence. The projection from the doubled-Ising elementary excitation state $\lvert \sigma\bar{\sigma}; 1, 1\rangle_{\text{DI}}$ to the toric-code state $\lvert m, 1, 1\rangle$

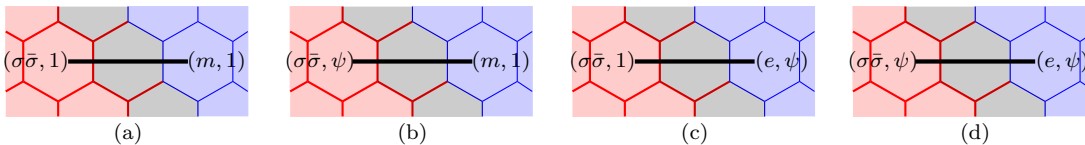

Figure 8: The interdomain elementary excitation states with doubled-Ising topological observable $\sigma\bar{\sigma}$.

corresponds to the allowed pair $\sigma\bar{\sigma}$-$m$ in the interdomain elementary excitation states

$$\left|\sigma\bar{\sigma}\text{-}m;1,1\right\rangle, \qquad \left|\sigma\bar{\sigma}\text{-}m;\psi,1\right\rangle, \tag{43}$$

while $P_{\text{eff}}\left|\sigma\bar{\sigma};\psi,\psi\right\rangle_{\text{DI}} = \left|e,\psi,\psi\right\rangle$ corresponds to the allowed pair $\sigma\bar{\sigma}$-$e$ in

$$\left|\sigma\bar{\sigma}\text{-}e;1,\psi\right\rangle, \qquad \left|\sigma\bar{\sigma}\text{-}e;\psi,\psi\right\rangle. \tag{44}$$

These four interdomain states all have $J_{\text{DI}} = \sigma\bar{\sigma}$, but $J_{\text{TC}}$ can be $m$ or $e$. See Fig. 8. This phenomenon is the spatial counterpart of the splitting of anyons in anyon condensation. Namely, an anyon $\sigma\bar{\sigma}$ in the doubled Ising domain may hop into the toric code domain by crossing the gapped domain wall and become either an anyon $e$ or $m$.

    With our model, splitting can also be understood dynamically as follows. The states $\left|\sigma\bar{\sigma};p,q\right\rangle_{\text{DI}}$ with $p,\,q \in \{1,\psi\}$ in Fig. 3 are indistinguishable in the doubled Ising phase, as they can be transformed into each other by the local operators $B^{1\sigma\psi\sigma}$ and $B^{\psi\sigma1\sigma}$ (111). The local operators $B_P^{pq\sigma}$ however do not commute with the condensation term $\Delta H$ (8) in Hamiltonian (9). After $\psi\bar{\psi}$ condensation, the charges $1$ and $\psi$ can no longer transform into each other by local operators, and are thus associated with individual topological observables $m$ and $e$ respectively. An infinite energy barrier $\Lambda \to \infty$ prevents the toric-code states $\left|m;1,1\right\rangle$ and $\left|e;\psi,\psi\right\rangle$ from transforming into each other.

## 5.2 Identification

Seen in Table 7, $\left|1\bar{1};1,1\right\rangle_{\text{DI}}$ and $\left|\psi\bar{\psi};1,1\right\rangle_{\text{DI}}$ in the parent phase are both projected to $\left|1;1,1\right\rangle$ in the toric code domain of our model, while $\left|\psi\bar{1};\psi,\psi\right\rangle_{\text{DI}}$ and $\left|1\bar{\psi};\psi,\psi\right\rangle_{\text{DI}}$ are both projected to $\left|\epsilon;\psi,\psi\right\rangle$. This phenomenon is called *identification* in anyon condensation.

    The projections from elementary excitation states $\left|1\bar{1};1,1\right\rangle_{\text{DI}}$ and $\left|\psi\bar{\psi};1,1\right\rangle_{\text{DI}}$ in the parent phase to $\left|1;1,1\right\rangle$ in our model individually correspond to the interdomain elementary excitation states

$$\left|1\bar{1}-1;1,1\right\rangle, \qquad \left|\psi\bar{\psi}-1;1,1\right\rangle, \tag{45}$$

which have different doubled-Ising topological obervables $1\bar{1}$ and $\psi\bar{\psi}$ but same toric-code topological obervable $1$. The projections from $\left|\psi\bar{1};\psi,\psi\right\rangle_{\text{DI}}$ and $\left|1\bar{\psi};\psi,\psi\right\rangle_{\text{DI}}$ to $\left|\epsilon;\psi,\psi\right\rangle$ respectively correspond to the interdomain elementary excitation states

$$\left|\psi\bar{1}-\epsilon;\psi,\psi\right\rangle, \qquad \left|1\bar{\psi}-\epsilon;\psi,\psi\right\rangle \tag{46}$$

with different doubled-Ising topological observables $\psi\bar{1}$ and $1\bar{\psi}$ but same toric-code topological observable $\epsilon$. It appears that the quasiparticles $(1\bar{1},1)$ and $(\psi\bar{\psi},1)$ in the doubled Ising domain become the same toric-code quasiparticle $(1,1)$ when hopping into the toric code domain, while $(\psi\bar{1},1)$ and $(1\bar{\psi},\epsilon)$ are identified to be $(\epsilon,\psi)$. This phenomenon is the spatial counterpart of identification in anyon condensation.

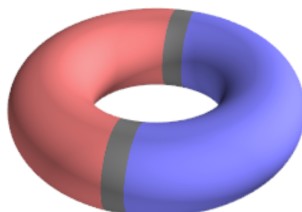

Figure 9: Our model on a torus: Two gapped domain walls (gray) separate the doubled Ising domain (red) and $\mathbb{Z}_2$ toric code domain (blue).

### 5.3    Confinements

Seen in Table 7, the states $\left|\sigma\bar{1};\sigma,\sigma\right\rangle_{\mathrm{DI}}$, $\left|\sigma\bar{\psi};\sigma,\sigma\right\rangle_{\mathrm{DI}}$, $\left|1\bar{\sigma};\sigma,\sigma\right\rangle_{\mathrm{DI}}$ and $\left|\psi\bar{\sigma};\sigma,\sigma\right\rangle_{\mathrm{DI}}$ in the parent phase are all projected to 0 in $\mathcal{H}_{\mathrm{eff}}$ of our model via $\psi\bar{\psi}$ condensation. This phenomenon is called *confinement* in anyon condensation. This is because in the toric code domain, the edges and tails cannot take value $\sigma$ in the states in $\mathcal{H}_{\mathrm{eff}}$.

Correspondingly, there is no interdomain elementary excitation states $\left|J_{\mathrm{DI}}\text{-}J_{\mathrm{TC}};p,q\right\rangle$ with $J_{\mathrm{DI}} = \sigma\bar{1}$, $\sigma\bar{\psi}$, $1\bar{\sigma}$ or $\psi\bar{\sigma}$, as the quasiparticles $(\sigma\bar{1},\sigma)$, $(\sigma\bar{\psi},\sigma)$, $(1\bar{\sigma},\sigma)$ and $(\psi\bar{\sigma},\sigma)$ in the doubled Ising domain cannot hop into the toric code domain unless overcoming infinite energy barriers $\Lambda \to \infty$. This phenomenon is the spatial counterpart of confinement in anyon condensation.

In anyon condensation, the doubled-Ising anyons $\sigma\bar{1}$ and $\sigma\bar{\psi}$ in the doubled Ising phase become the same quasiparticle $\chi$ in the intermediate phase, and $1\bar{\sigma}$ and $\psi\bar{\sigma}$ become $\bar{\chi}$; however, $\chi$ and $\bar{\chi}$ in the intermediate phase are confined in the $\mathbb{Z}_2$ toric code phase because of their nontrivial braiding with the new vacuum in the intermediate phase. Now that the gapped domain wall is the spatial counterpart of the intermediate phase, we can see that domainwall quasiparticles $\chi$ and $\bar{\chi}$ also have nontrivial braiding with the trivial quasiparticle 1 in the gapped domain wall (to be defined in Eq. (66)).

## 6    The bases of the ground states on the torus

The defining properties of a topological phase are usually obtained from the ground states of the topological phase on the torus [24, 27, 28]. For example, on the torus, a topological phase has a ground-state degeneracy, which is a topological quantum number of the topological phase. For instances, on the torus, the doubled Ising phase has GSD = 9, while the $\mathbb{Z}_2$ toric code phase has GSD = 4. In this section, we shall find two distinct and typical ground-state bases of our model on the torus (see Fig. 9), using noncontractible loop operators to be constructed shortly. These two ground-state bases will lead us to the characteristic properties of our model, as to be shown in Sections 6.3, 7.1 and 7.2.

### 6.1    The domainwall basis of the ground-state subspace

Sewing the two ends of a ribbon operator results in a loop operator [24, 27]. If the loop path is noncontractible, we have a *noncontractible loop operator*. Loop operators preserve the ground-state subspace because no anyons are created. On the torus, there are two homotopic classes of noncontractible loops: $V$ loop along the gapped domain wall, and $H$ loop across the gapped domain wall. Here $V$ stands for "vertical" and $H$ "horizontal". See Fig. 10.

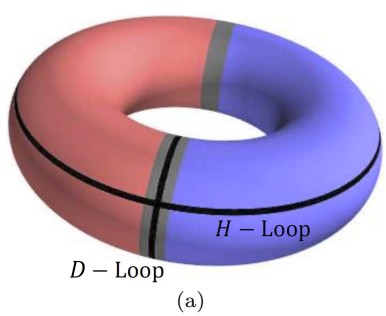 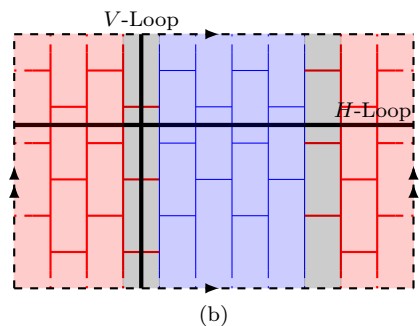

Figure 10: (a) Gapped domain walls and noncontractible loops on the torus. (b) The corresponding lattice picture of (a).

There are 6 loop operators $W_H^{J_{\mathrm{DI}}\text{-}J_{\mathrm{TC}}}$ along $H$-loop, labeled by the interdomain topological observable pairs $J_{\mathrm{DI}}\text{-}J_{\mathrm{TC}}$:

$$W_H^{1\bar{1}\text{-}1} = I, \qquad W_H^{\psi\bar{\psi}\text{-}1}, \qquad W_H^{\psi\bar{1}\text{-}\epsilon}, \qquad W_H^{1\bar{\psi}\text{-}\epsilon}, \qquad W_H^{\sigma\bar{\sigma}\text{-}m}, \qquad W_H^{\sigma\bar{\sigma}\text{-}e}. \tag{47}$$

Similarly, there are 6 noncontratible loop operators along the $V$-loop, labled by $J_{\mathrm{DW}}$:

$$W_V^1 = I, \qquad W_V^\epsilon, \qquad W_V^m, \qquad W_V^e, \qquad W_V^{\bar{\chi}}, \qquad W_V^\chi. \tag{48}$$

All these operators are linearly independent. They generate an algebra denoted by $\mathcal{A}$. See Appendix F.1.

The algebra $\mathcal{A}$ generates the entire ground-state subspace $\mathcal{H}_0$ of our model given any ground state:

$$\mathcal{H}_0 = \mathcal{A}\,|\Phi\rangle, \quad \forall\,|\Phi\rangle \in \mathcal{H}_0\backslash\{0\}. \tag{49}$$

We leave the full proof of Eq. (49) in Appendix F.2 but sketch the proof as follows. Since $P_{\mathrm{eff}}$ commutes with the loop operators of the parent phase in $\mathcal{H}_{\mathrm{eff}}$:

$$P_{\mathrm{eff}}\left[W_H^{J_{\mathrm{DI}}}, P_{\mathrm{eff}}\right] = P_{\mathrm{eff}}\left[W_V^{J_{\mathrm{DI}}}, P_{\mathrm{eff}}\right] = 0, \tag{50}$$

it projects the doubled-Ising loop operators $W_H^{J_{\mathrm{DI}}}$ and $W_V^{J_{\mathrm{DI}}}$ to $W_H^{J_{\mathrm{DI}}\text{-}J_{\mathrm{TC}}}$ and $W_V^{J_{\mathrm{DW}}}$ of our model, and projects the ground-state subspace $\mathcal{H}_0^{\mathrm{DI}}$ of the parent phase to $\mathcal{H}_0$ of our model. Now that the doubled-Ising loop operators $W_H^{J_{\mathrm{DI}}}$ and $W_V^{J_{\mathrm{DI}}}$ can generate $\mathcal{H}_0^{\mathrm{DI}}$ given any ground state of the parent phase, $\mathcal{A}$ can generate $\mathcal{H}_0$ of our model given any ground state of our model.

We shall construct a ground-state basis using $W_V^{J_{\mathrm{DW}}}$ as follows. There exists a unique ground state $|\Phi\rangle_V \in \mathcal{H}_0$, such that

$$W_H^{J_{\mathrm{DI}}\text{-}J_{\mathrm{TC}}}\,|\Phi\rangle_V = d_{J_{\mathrm{DI}}\text{-}J_{\mathrm{TC}}}\,|\Phi\rangle_V \tag{51}$$

for all operators $W_H^{J_{\mathrm{DI}}\text{-}J_{\mathrm{TC}}}$, which generate a largest commutative subalgebra of $\mathcal{A}$. Here $d_{J_{\mathrm{DI}}\text{-}J_{\mathrm{TC}}} = 1$ for all pairs $J_{\mathrm{DI}}\text{-}J_{\mathrm{TC}}$ are the only positive eigenvalues of $W_H^{J_{\mathrm{DI}}\text{-}J_{\mathrm{TC}}}$. This common eigenstate can be obtained up to factors by

$$|\Phi\rangle_V = P_H\,|\varphi\rangle \tag{52}$$

given arbitrary $|\varphi\rangle \in \mathcal{H}_{\mathrm{eff}}$, where

$$P_H = \frac{I + W_H^{\psi\bar{\psi}\text{-}1}}{2}\,\frac{I + W_H^{\psi\bar{1}\text{-}\epsilon}}{2}\,\frac{I + W_H^{1\bar{\psi}\text{-}\epsilon}}{2}\,\frac{I + W_H^{\psi\bar{\psi}\text{-}1} + 2W_H^{\sigma\bar{\sigma}\text{-}m}}{4}\,\times$$

$$\frac{I + W_H^{\psi\bar{\psi}\text{-}1} + 2W_H^{\sigma\bar{\sigma}\text{-}e}}{4} \, P_0. \tag{53}$$

Here,

$$P_0 = \prod_{P \in \text{DI}} B_P^{\text{DI}} \prod_{P \in \text{DW}} B_P^{\text{DW}} \prod_{P \in \text{TC}} B_P^{\text{TC}} \prod_V Q_V, \tag{54}$$

such that $P_0 \mathcal{H}_{\text{eff}} = \mathcal{H}_0$, where $B_P^{\text{DI}}$, $B_P^{\text{DW}}$, $B_P^{\text{TC}}$ and $Q_V$ are the plaquette operators and vertex operators in the Hamiltonian $H_{\text{eff}}$ (16) of our model.

Since $|\Phi\rangle_V$ is the common eigenstate of all $W_H^{J_{\text{DI}}\text{-}J_{\text{TC}}}$, according to Eq. (49),

$$\mathcal{H}_0 = \text{span}\left\{ W_V^{J_{\text{DW}}} |\Phi\rangle_V \right\}. \tag{55}$$

The states $W_V^{J_{\text{DW}}} |\Phi\rangle_V$ are orthonormal and thus form a basis of $\mathcal{H}_0$. We define

$$|J_{\text{DW}}\rangle_V := W_V^{J_{\text{DW}}} |\Phi\rangle_V. \tag{56}$$

We call this basis the *domainwall basis*, depicted in Fig. 11a.

## 6.2 The interdomain basis of the ground-state subspace

The algebra $\mathcal{A}$ has more than one largest commutative subalgebra. The 6 $V$-loop operators $W_V^{J_{\text{DW}}}$ also generate a largest commutative subalgebra of $\mathcal{A}$ and determine another unique ground state $|\Phi\rangle_H$, such that

$$W_V^{J_{\text{DW}}} |\Phi\rangle_H = d_{J_{\text{DW}}} |\Phi\rangle_H, \tag{57}$$

where $d_1 = d_\epsilon = d_m = d_e = 1$, $d_\chi = d_{\bar{\chi}} = \sqrt{2}$. This common eigenstate can be obtained up to factors as

$$|\Phi\rangle_H = P_V |\varphi\rangle, \qquad \forall |\varphi\rangle \in \mathcal{H}_{\text{eff}}, \tag{58}$$

where

$$P_V = \frac{I + W_V^\epsilon}{2} \frac{I + W_V^m}{2} \frac{I + W_V^e}{2} \frac{I + W_V^\epsilon + \sqrt{2}W_V^{\bar{\chi}}}{4} \frac{I + W_V^\epsilon + \sqrt{2}W_V^\chi}{4} \, P_0. \tag{59}$$

Hence, we obtain what we call the *interdomain basis* of $\mathcal{H}_0$:

$$|J_{\text{DI}} - J_{\text{TC}}\rangle_H := W_H^{J_{\text{DI}}-J_{\text{TC}}} |\Phi\rangle_H, \tag{60}$$

as depicted in Fig. 11b.

## 6.3 Ground-state degeneracy on the torus

According to the domainwall basis $|J_{\text{DW}}\rangle_V$ (56) or interdomain basis $|J_{\text{DI}}\text{-}J_{\text{TC}}\rangle_H$ (60) of the ground-state subspace on the torus, our model of the doubled Ising and $\mathbb{Z}_2$ toric code phases separated by two gapped domain walls on the torus has

$$\text{GSD}_{\text{torus}} = 6. \tag{61}$$

This GSD agrees with the number of the domainwall quasiparticle species $J_{\text{DW}}$, as well as the number of interdomain topological observable pairs $J_{\text{DI}}\text{-}J_{\text{TC}}$. This is a generalization of the correspondence between the GSD of a topological phase on the torus and the number of anyon species of this topological phase. We can simply replace the input data of our

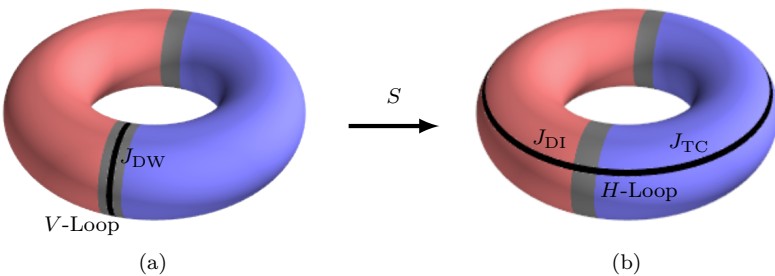

Figure 11: The $S$ matrix: the basis transformation from (a) the domainwall basis $\{|J_{\mathrm{DW}}\rangle_V\}$ to (b) the interdomain basis $\{|J_{\mathrm{DI}} - J_{\mathrm{TC}}\rangle_H\}$.

model with that of any other parent and child phases: for any two domain-wall-separated topological phases related by anyon condensation, the following correspondence holds.

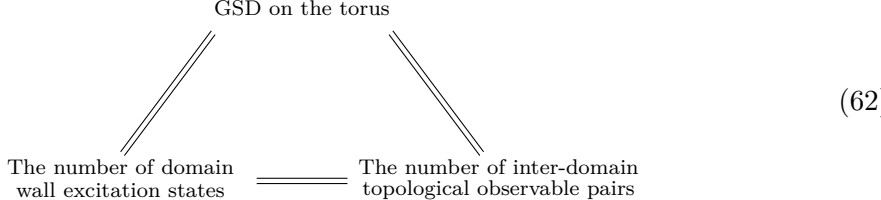

$$\tag{62}$$

Note that the GSD in Eq. (61) has also been obtained before by algebraic methods [16].

# 7   The $S$ and $T$ matrices

Besides the ground-state degeneracy, another fingerprint of the topological phase consists of the $S$ and $T$ matrices, which generate the basis transformations in the ground-state Hilbert space $\mathcal{H}_0$ on the torus. We shall construct the $S$ and $T$ matrices of our model and show their physical significance.

## 7.1   The $S$ matrix on the torus

We define the $S$ matrix as the basis transformation (see Fig. 11):

$$S_{J_{\mathrm{DW}}, J_{\mathrm{DI}}-J_{\mathrm{TC}}} := {}_V\langle J_{\mathrm{DW}}|J_{\mathrm{DI}} - J_{\mathrm{TC}}\rangle_H \,, \tag{63}$$

which up to phase factors reads

$$
S = \frac{\sqrt{2}}{4}
\begin{array}{c}
\\ 1 \\ \epsilon \\ m \\ e \\ \bar{\chi} \\ \chi
\end{array}
\begin{array}{cccccc}
1\bar{1}-1 & \psi\bar{\psi}-1 & \psi\bar{1}-\epsilon & 1\bar{\psi}-\epsilon & \sigma\bar{\sigma}-m & \sigma\bar{\sigma}-e \\
\left(\begin{array}{cccccc}
1 & 1 & 1 & 1 & 1 & 1 \\
1 & 1 & 1 & 1 & -1 & -1 \\
1 & 1 & -1 & -1 & 1 & -1 \\
1 & 1 & -1 & -1 & -1 & 1 \\
\sqrt{2} & -\sqrt{2} & \sqrt{2} & -\sqrt{2} & 0 & 0 \\
\sqrt{2} & -\sqrt{2} & -\sqrt{2} & \sqrt{2} & 0 & 0
\end{array}\right)
\end{array}. \tag{64}
$$

The Levin-Wen model of a single topological phase on the torus is invariant under rotations generated by a $\frac{\pi}{2}$ rotation of the lattice [31]; the $S$ matrix of the model represents the $\frac{\pi}{2}$ rotation and is thus symmetric and unitary. Nevertheless, our model on the torus does not have this rotation invariance due to the gapped domain walls, so the $S$ matrix (63) has nothing to do with rotations. Since the domainwall basis and interdomain basis are labeled by different sets of quasiparticle species, our $S$ matrix is neither symmetric nor unitary.

The $S$ matrix of a single topological phase not only transforms the ground-state bases on the torus but also characterizes the mutual statistics of the anyons in the topological phase. This feature of the $S$ matrix is generalized in our model. That is, the matrix elements of our $S$ matrix (64) can be understood in the following sense of braiding

$$
\left| \begin{matrix} J_{\mathrm{DI}} & & J_{\mathrm{TC}} \\ & J_{\mathrm{DW}} & \end{matrix} \right\rangle = \frac{S_{J_{\mathrm{DW}}, J_{\mathrm{DI}} - J_{\mathrm{TC}}}}{S_{1, 1\bar{1}-1}} \left| \phantom{xxxx} \right\rangle . \tag{65}
$$

Note that in Eq. (64), domainwall quasiparticles $\chi$ and $\bar{\chi}$ have nontrivial mutual statistics with the trivial domainwall quasiparticle 1:

$$
\frac{S_{\chi, \psi\bar{\psi}-1}}{S_{1, 1\bar{1}-1}} = \frac{S_{\bar{\chi}, \psi\bar{\psi}-1}}{S_{1, 1\bar{1}-1}} = -\sqrt{2}. \tag{66}
$$

See Section 5.3.

## 7.2 The $T$ matrix on the torus

Although our model on the torus is not invariant under the $\frac{\pi}{2}$ rotation of the lattice, it is still invariant under the shear, i.e., the $\mathcal{T}$ transformation, of the lattice along the vertical direction.

The $\mathcal{T}$ transformation exchanges the positions of two vertically neighboring vertices in the ovals in Fig. 12. After the $\mathcal{T}$ transformation, for any two horizontally adjacent plaquettes in the original lattice, the one on the right is shifted by one plaquette upward relative to the one on the left. See for example the plaquettes $P_1$ and $P_2$ in Fig. 12b.

Figure 13 shows how the $\mathcal{T}$ transformation acts on the entire lattice. In the lattice, the number of columns is equal to the number of plaquettes in each column (e.g., the number is 4 for the lattice in Fig. 13), so the configuration of the lattice on the torus is invariant and thus the Hilbert space of our model is unchanged under the $\mathcal{T}$ transformation. The $\mathcal{T}$ transformation can be represented in this invariant Hilbert space. Note that any loop remains a loop under the $\mathcal{T}$ transformation; hence, the $\mathcal{T}$ transformation preserves the ground-state Hilbert space $\mathcal{H}_0$.

To see how the $\mathcal{T}$ transformation acts on a basis state of the Hilbert space, we can zoom in to see how $\mathcal{T}$ acts in the vicinity of a dashed oval:

$$
\left| \begin{matrix} & a_0 \\ e_1 & a_1 \\ & e_2 \\ & a_2 \end{matrix} \right\rangle \implies \sqrt{d_{a_1} d_{a_1'}}\, G^{a_0 e_1 a_1}_{a_2 e_2 a_1'} \left| \begin{matrix} & a_0 \\ e_1 & a_1' \\ & e_2 \\ & a_2 \end{matrix} \right\rangle . \tag{67}
$$

We define the $T$ matrix as a representation of the $\mathcal{T}$ transformation over the interdomain basis:

$$
T_{J_{\mathrm{DW}} - J_{\mathrm{TC}}, K_{\mathrm{DW}} - K_{\mathrm{TC}}} := {}_H\langle J_{\mathrm{DI}} - J_{\mathrm{TC}}| \, \mathcal{T} \, |K_{\mathrm{DW}} - K_{\mathrm{TC}}\rangle_H , \tag{68}
$$

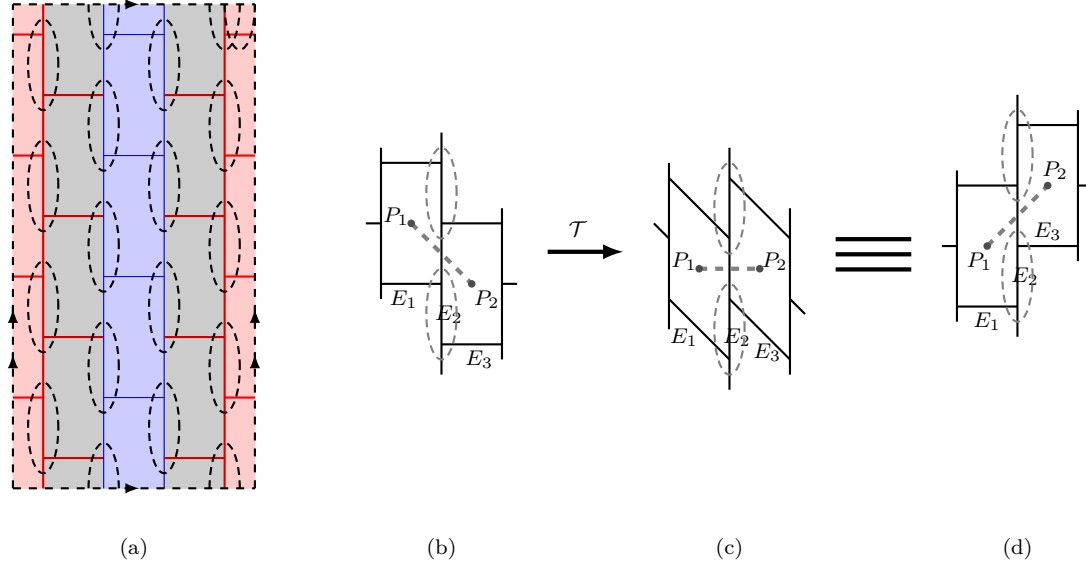

|  (a)  |  (b)  |  (c)  |  (d)  |

Figure 12: The $\mathcal{T}$ transformation and change of perspectives. (a) The original lattice on the torus. Each oval encircles the vertices to be exchanged under the $\mathcal{T}$ transformation. (b) Two horizontally adjacent plaquettes $P_1$ and $P_2$. (c) The plaquettes $P_1$ and $P_2$ after the $\mathcal{T}$ transformation. (d) is (c) in a more convenient perspective.

The matrix $T$ is diagonal and reads

$$
\begin{array}{c|cccccc}
J_{\mathrm{DI}} - J_{\mathrm{TC}} & 1\bar{1} - 1 & \psi\bar{\psi} - 1 & \psi\bar{1} - \epsilon & 1\bar{\psi} - \epsilon & \sigma\bar{\sigma} - m & \sigma\bar{\sigma} - e \\
\hline
T_{J_{\mathrm{DI}}-J_{\mathrm{TC}},J_{\mathrm{DI}}-J_{\mathrm{TC}}} & 1 & 1 & -1 & -1 & 1 & 1
\end{array}
\tag{69}
$$

The diagonal elements $T_{J_{\mathrm{DI}}-J_{\mathrm{TC}},J_{\mathrm{DI}}-J_{\mathrm{TC}}}$ of the $T$ matrix on the torus are also the topological spins $\theta_{J_{\mathrm{DI}}} = \theta_{J_{\mathrm{TC}}} = T_{J_{\mathrm{DI}}-J_{\mathrm{TC}},J_{\mathrm{DI}}-J_{\mathrm{TC}}}$ of the anyons $J_{\mathrm{DI}}$ and $J_{\mathrm{TC}}$:

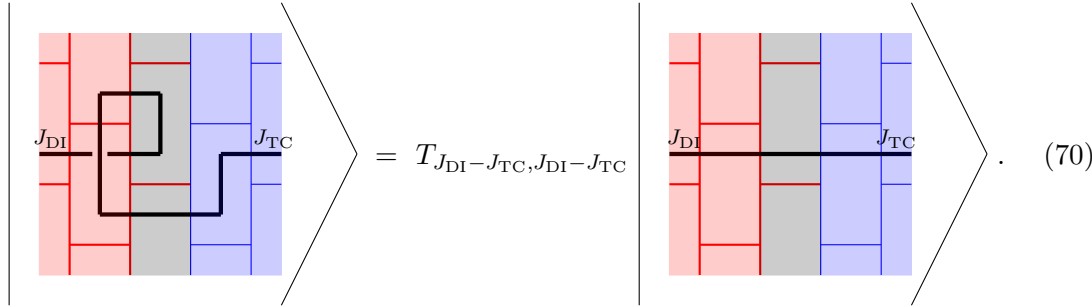

$$
= T_{J_{\mathrm{DI}}-J_{\mathrm{TC}},J_{\mathrm{DI}}-J_{\mathrm{TC}}} \qquad . \tag{70}
$$

The $S$ matrix (63) and $T$ matrix (68) generate all possible basis transformations of the ground states of our model on the torus.

# 8   Conclusion

In this paper, we begin with an extended LW model describing a parent phase and trigger the anyon condensation in half of the system to construct an exactly solvable lattice model describing the parent phase and its child phase separated by a gapped domain wall. To make various properties of the model specific and explicit, we focus on the model in the

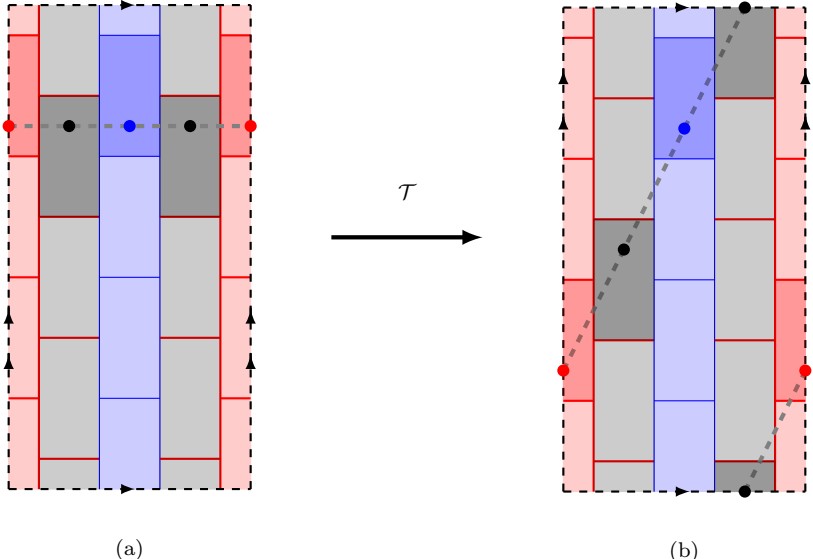

Figure 13: The $\mathcal{T}$ transformation on the entire torus. The number of columns and the number of plaquettes in each column are both 4. (a) The original lattice. (b) The lattice after the $\mathcal{T}$ transformation and the change of perspective. The plaquettes in deeper colors illustrate how the plaquettes shift under the $\mathcal{T}$ transformation.

case of the doubled Ising and $\mathbb{Z}_2$ toric code topological phases with a gapped domain wall in between. Our model leads to the following main results.

1. We obtain the spectrum of the elementary excitation states of the system.

2. We explicitly show the correspondence between the transformations of anyons in anyon condensation with the elementary excitation states in our model.

3. We find the ground-state bases of our model on the torus, and show that the ground-state degeneracy (GSD) on the torus equals the number of quasiparticle species in the gapped domain wall.

4. We construct the $S$ and $T$ matrices that generate the basis transformations of the ground states on the torus.

## Acknowledgements

YW is supported by NSFC grant No. 11875109, General Program of Science and Technology of Shanghai No. 21ZR1406700, Fudan University Original Project (Grant No. IDH1512092/009), and Shanghai Municipal Science and Technology Major Project (Grant No.2019SHZDZX01). YW is grateful to the Hospitality of the Perimeter Insitute during his visit, where the main part of this work is done.

## A Gauge transformations of the positions of tails

In the lattice of the extended LW model, each tail associated with vertex $V$ is chosen to attach to any one of the three edges incident at $V$. Different choices lead to different lattice configurations and hence different Hilbert spaces of the extended LW model. Nevertheless, since tails are internal degrees of freedom that cannot be probed, the different Hilbert spaces underline the same topological phase. Specifically, these Hilbert spaces are equivalent up to the gauge transformation $\mu$:

$$\mu_p \left| \begin{array}{c} V \\ i \quad j \\ l \\ k \quad p \end{array} \right\rangle = \sum_{m \in L_{\mathrm{DI}}} \sqrt{d_l d_m} \; G_{jim}^{kpl} \left| \begin{array}{c} V \\ i \quad m \quad j \\ k \quad p \end{array} \right\rangle . \tag{71}$$

Besides the gauge transformation of the positions of tails, the directions of tails are also defined up to gauge transformations [26]. For example, the following two states are equivalent up to a gauge transformation:

$$\left| \begin{array}{c} V \\ i \quad j \\ l \\ k \quad p \end{array} \right\rangle \;\equiv\; \left| \begin{array}{c} V \\ i \quad j \\ l \\ p \quad k \end{array} \right\rangle . \tag{72}$$

## B The matrix elements of the doubled-Ising ribbon operators

In Section 4.1, we have defined the action of shortest ribbon operators $W_E^{J_{\mathrm{DI}};p,q}$ on the states where all tails take value $1 \in L_{\mathrm{DI}}$:

$$W_E^{J_{\mathrm{DI}};p,q} \left| \begin{array}{c} j_E \end{array} \right\rangle = \sum_{k \in L_{\mathrm{DI}}} \sqrt{\frac{d_k}{d_{j_E}}} \, z_{pqj_E}^{J_{\mathrm{DI}};k} \left| \begin{array}{c} j_E \quad q \\ p \quad k \\ j_E \end{array} \right\rangle , \tag{73}$$

where $j_E \in L_{\mathrm{DI}}$ is the label on edge $E$, and $z_{pqj_E}^{J_{\mathrm{DI}};k}$ are listed in Appendix D.1.

In this appendix, we define the actions of any ribbon operators on any states in $\mathcal{H}$ of the doubled Ising phase.

### B.1 Matrix elements of shortest ribbon operators

Here, we define the action of the shortest ribbon operator $W_E^{J_{\mathrm{DI}};p,q}$ on the states, in which there are nontrivial tails attached on edge $E$.

We start with the simplest case where the tail is associated with the upper vertex $V_2$ of edge $E$ and carries a charge $r$ and points to the right:

$$W_E^{J_{\mathrm{DI}};p,q} \left| \begin{array}{c} V_2 \\ j_2 \quad r \\ j_0 \end{array} \right\rangle = \sum_{ks \in L_{\mathrm{DI}}} \sqrt{d_k d_s} \, z_{pqj_0}^{J_{\mathrm{DI}},k} \, G_{kqs}^{rj_2j_0} \left| \begin{array}{c} V_2 \\ j_2 \quad s \\ p \quad k \\ j_0 \end{array} \right\rangle . \tag{74}$$

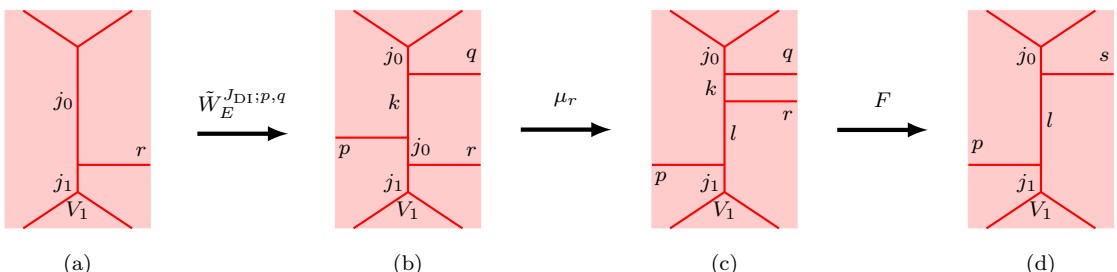

Figure 14: The action of $W_E^{J_{\mathrm{DI}};p,q}$ on the state where a tail $r$ is associated with $V_1$.

This action formally is the composition of two operators $\tilde{W}_E^{J_{\mathrm{DI}};p,q}$ and $F$:

$$W_E^{J_{\mathrm{DI}};p,q} \left| \vphantom{\begin{array}{c}V_2\\j_2\\j_0\end{array}} \right. \left. \text{(diagram)} \right\rangle = F\tilde{W}_E^{J_{\mathrm{DI}};p,q} \left| \text{(diagram)} \right\rangle . \tag{75}$$

First, the operator $\tilde{W}_E^{J_{\mathrm{DI}};p,q}$ acts as

$$\tilde{W}_E^{J_{\mathrm{DI}};p,q} \left| \text{(diagram)} \right\rangle = \sum_{k \in L_{\mathrm{DI}}} \sqrt{\frac{d_k}{d_{j_0}}}\, z_{pqj_0}^{J_{\mathrm{DI}};k} \left| \text{(diagram)} \right\rangle . \tag{76}$$

Now there are two tails ($q$ and $r$) associated with vertex $V_2$ on edge $E$, which can then be fused by operator $F$:

$$F \left| \text{(diagram)} \right\rangle := \sum_{s \in L_{\mathrm{DI}}} \sqrt{d_{j_0} d_s}\, G_{kqs}^{r j_2 j_0} \left| \text{(diagram)} \right\rangle . \tag{77}$$

The result is Eq. (74).

Similarly, when edge $E$ has one tail ($r$) associated with the lower vertex $V_1$ and pointing right, $W_E^{J_{\mathrm{DI}};p,q}$ acts on the state as

$$W_E^{J_{\mathrm{DI}};p,q} \left| \text{(diagram)} \right\rangle = \sum_{kls \in L_{\mathrm{DI}}} d_k \sqrt{d_l d_s}\, z_{pqj_0}^{J_{\mathrm{DI}},k} G_{j_1 rl}^{kpj_0} G_{lrs}^{qj_0 k} \left| \text{(diagram)} \right\rangle , \tag{78}$$

which is also a composition of the actions of two operators:

$$W_E^{J_{\mathrm{DI}};p,q} \left| \text{(diagram)} \right\rangle = F\mu_r \tilde{W}_E^{J_{\mathrm{DI}};p,q} \left| \text{(diagram)} \right\rangle . \tag{79}$$

See Fig. 14.

All other matrix elements of ribbon operators $W_E^{J_{\mathrm{DI}};p,q}$ can be obtained likewise.

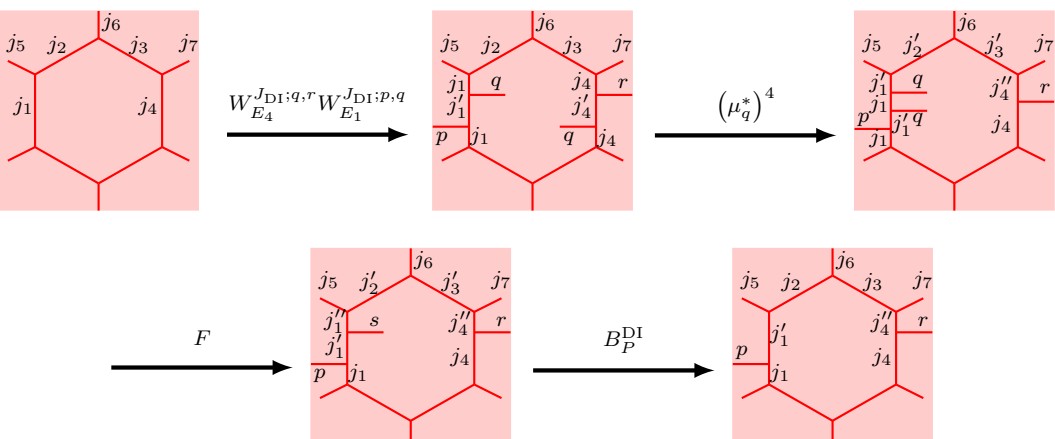

Figure 15: Concatenating shorter ribbon operators to a longer one.

## B.2 Concatenating shorter ribbon operators to a longer ribbon operator

Now we define the ribbon operators along longer paths. Consider ribbon operator $W_L^{J_{\mathrm{DI}};p,q}$ in Fig. 4c, whose path $L$ crosses two edges labeled by $j_1$ and $j_4$ respectively:

$$
W_L^{J_{\mathrm{DI}};p,r} \left| \quad \right\rangle = \sum_{q j_1' j_2' j_3' j_4' j_4'' \in L_{\mathrm{DI}}} \overline{z_{pqj_1}^{J_{\mathrm{DI}};j_1'}} \; \overline{z_{qrj_4}^{J_{\mathrm{DI}};j_4'}} \times
$$

$$
\sqrt{\frac{d_{j_1}}{d_{j_1'}}} \; G_{qj_4j_4''}^{rj_4j_4'} \; G_{qj_4'j_3'}^{j_7j_3j_4} \; G_{qj_3'j_2'}^{j_6j_2j_3} \; G_{qj_2'j_1'}^{j_5j_1j_2} \left| \quad \right\rangle . \tag{80}
$$

The operator $W_L^{J_{\mathrm{DI}};p,r}$ can be formally written as

$$
W_L^{J_{\mathrm{DI}};p,r} = B_P^{\mathrm{DI}} \sum_{q \in L_{\mathrm{DI}}} F(\mu_q^*)^4 W_{E_4}^{J_{\mathrm{DI}};q,r} W_{E_1}^{J_{\mathrm{DI}};p,q}, \tag{81}
$$

see Fig. 15.

All matrix elements of the doubled-Ising ribbon operators taking any paths can be obtained likewise.

## C Proof of the commutation in $\mathcal{H}_{\mathrm{eff}}$ of $P_{\mathrm{eff}}$ and the doubled-Ising ribbon operators

In this section we prove Eq. (24):

$$
P_{\mathrm{eff}} \left[ W_L^{J_{\mathrm{DI}};p,q}, P_{\mathrm{eff}} \right] = 0. \tag{82}
$$

Obviously, Eq. (24) holds when $J_{\mathrm{DI}} = 1\bar{1}$, $\psi\bar{\psi}$, $\psi\bar{1}$, $1\bar{\psi}$ and $\sigma\bar{\sigma}$ because these anyon species all have charges in $\{1, \psi\}$ that are preserved under the projection.

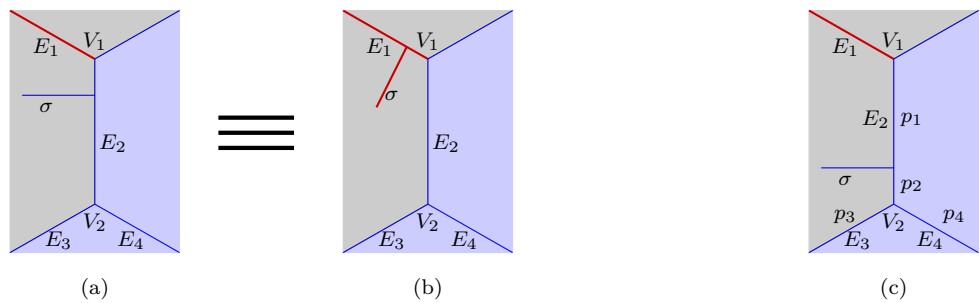

Figure 16: Two different cases where a tail with charge $\sigma$ is attached to a TC edge $E_2$. (a) The tail $\sigma$ is associated with the vertex $V_1$ with an incident DI edge $E_1$. (b) is equivalent to (a) up to a $\mu_\sigma^*$ gauge transformation. (c) The tail $\sigma$ is associated with the vertex $V_2$ with two incident TC edges $E_3$ and $E_4$.

We then consider the ribbon operators $W_L^{J_{\mathrm{DI}};p,q}$ with $J_{\mathrm{DI}} = \sigma\bar{1}$, $\sigma\bar{\psi}$, $1\bar{\sigma}$ and $\psi\bar{\sigma}$. These operators create quasiparticles with charge $\sigma$ at the ends of path $L$. Note that Eq. (24) holds when path $L$ crosses DI edges because $P_{\mathrm{eff}}$ only acts on the TC edges, we only need to consider the projections of states $|\varphi\rangle = W_L^{J_{\mathrm{DI}};p,q}|\Phi\rangle$ with nontrivial tails $\sigma$ on TC edges. There are two cases of these states, depicted in Fig. 16a and Fig. 16c.

In the first case, the tail $\sigma$ is associated with vertex $V_1$ with an incident DI edge $E_1$. Up to the gauge transformations introduced in Appendix A, this state is equivalent to the state with tail $\sigma$ on DI edge $E_1$ (see Fig. 16b) and therefore satisfies Eq. (24).

In the second case, the tail $\sigma$ is associated with vertex $V_2$ with two other incident TC edges $E_3$ and $E_4$. Note that

$$N_{\sigma\sigma}^1 = N_{\sigma\sigma}^\psi = 1, \qquad N_{11}^\sigma = N_{1\psi}^\sigma = N_{\psi1}^\sigma = N_{\psi\psi}^\sigma = 0, \tag{83}$$

one of the labels $p_1$ and $p_2$ on edge $E_2$ must be $\sigma$. If $p_1 = \sigma$, $W_{E_2}^{\psi\bar{\psi};1,1}|\varphi\rangle = -|\varphi\rangle$; otherwise, if $p_2 = \sigma$, one of the labels $p_3$ and $p_4$ must be $\sigma$. Since associated with vertex $V$ there is at most one nontrivial tail that has been on edge $E_2$, we have $W_{E_3}^{\psi\bar{\psi};1,1} = -|\varphi\rangle$ or $W_{E_4}^{\psi\bar{\psi};1,1}|\varphi\rangle = -|\varphi\rangle$. Therefore,

$$P_{\mathrm{eff}}|\varphi\rangle = 0, \tag{84}$$

which leads to

$$P_{\mathrm{eff}}W_L^{J_{\mathrm{DI}};p,q}P_{\mathrm{eff}} = P_{\mathrm{eff}}W_L^{J_{\mathrm{DI}};p,q} = 0. \tag{85}$$

# D   The components of $z$ tensors

## D.1   Nonzero components of $z^{J_{\mathrm{DI}}}$ tensors in the doubled Ising domain

Equation (22)

$$\frac{\delta_{j,t}N_{rs}^t}{d_t}z_{pqt}^{J_{\mathrm{DI}};w} = \sum_{ulv \in L_{\mathrm{DI}}} d_u d_v z_{lqr}^{J_{\mathrm{DI}};v} z_{pls}^{J_{\mathrm{DI}};u} G_{pwu}^{rst} G_{qwv}^{srj} G_{rvw}^{sul} \tag{86}$$

has 9 minimal solutions $z^{J_{\mathrm{DI}}}$, labeled by the 9 double-Ising anyon species. The nonzero components of these tensors are

$$z_{111}^{1\bar{1},1} = z_{11\psi}^{1\bar{1},\psi} = z_{11\sigma}^{1\bar{1},\sigma} = 1, \tag{87}$$

$$z_{111}^{\psi\bar{\psi},1} = z_{11\psi}^{\psi\bar{\psi},\psi} = 1, \qquad z_{11\sigma}^{\psi\bar{\psi},\sigma} = -1, \tag{88}$$

$$z_{\psi\psi 1}^{\psi\bar{1},\psi} = 1, \qquad z_{\psi\psi\psi}^{\psi\bar{1},1} = -1, \qquad z_{\psi\psi\sigma}^{\psi\bar{1},\sigma} = i, \tag{89}$$

$$z_{\psi\psi 1}^{1\bar{\psi},\psi} = 1, \qquad z_{\psi\psi\psi}^{1\bar{\psi},1} = -1, \qquad z_{\psi\psi\sigma}^{1\bar{\psi},\sigma} = -i, \tag{90}$$

$$z_{111}^{\sigma\bar{\sigma},1} = z_{\psi\psi 1}^{\sigma\bar{\sigma},\psi} = z_{\psi\psi\psi}^{\sigma\bar{\sigma},1} = 1, \qquad z_{11\psi}^{\sigma\bar{\sigma},\psi} = -1, \qquad z_{1\psi\sigma}^{\sigma\bar{\sigma},\sigma} = z_{\psi 1\sigma}^{\sigma\bar{\sigma},\sigma} = 1, \tag{91}$$

$$z_{\sigma\sigma 1}^{\sigma\bar{1},\sigma} = 1, \qquad z_{\sigma\sigma\psi}^{\sigma\bar{1},\sigma} = i, \qquad z_{\sigma\sigma\sigma}^{\sigma\bar{1},1} = e^{\frac{i\pi}{8}}, \qquad z_{\sigma\sigma\sigma}^{\sigma\bar{1},\psi} = e^{-\frac{3i\pi}{8}}, \tag{92}$$

$$z_{\sigma\sigma 1}^{\sigma\bar{\psi},\sigma} = 1, \qquad z_{\sigma\sigma\psi}^{\sigma\bar{\psi},\sigma} = i, \qquad z_{\sigma\sigma\sigma}^{\sigma\bar{\psi},1} = e^{-\frac{7i\pi}{8}}, \qquad z_{\sigma\sigma\sigma}^{\sigma\bar{\psi},\psi} = e^{\frac{5i\pi}{8}}, \tag{93}$$

$$z_{\sigma\sigma 1}^{1\bar{\sigma},\sigma} = 1, \qquad z_{\sigma\sigma\psi}^{1\bar{\sigma},\sigma} = -i, \qquad z_{\sigma\sigma\sigma}^{1\bar{\sigma},1} = e^{-\frac{i\pi}{8}}, \qquad z_{\sigma\sigma\sigma}^{1\bar{\sigma},\psi} = e^{\frac{3i\pi}{8}}, \tag{94}$$

$$z_{\sigma\sigma 1}^{\psi\bar{\sigma},\sigma} = 1, \qquad z_{\sigma\sigma\psi}^{\psi\bar{\sigma},\sigma} = -i, \qquad z_{\sigma\sigma\sigma}^{\psi\bar{\sigma},1} = e^{\frac{7i\pi}{8}}, \qquad z_{\sigma\sigma\sigma}^{\psi\bar{\sigma},\psi} = e^{-\frac{5i\pi}{8}}. \tag{95}$$

## D.2 Nonzero components of $z^{J_{\mathrm{TC}}}$ tensors in the toric code domain

The tensors $z^{J_{\mathrm{TC}}}$, $J_{\mathrm{TC}} \in \{1, e, m, \epsilon\}$, are

$$\begin{aligned} z_{pqs}^{1,u} = z_{pqs}^{1\bar{1},u} = z_{pqs}^{\psi\bar{\psi},u}, \qquad & z_{pqs}^{\epsilon,u} = z_{pqs}^{\psi\bar{1},u} = z_{pqs}^{1\bar{\psi},u}, \qquad z_{pqs}^{m,u} = \delta_{p,1} z_{pqs}^{\sigma\bar{\sigma},u}, \\ z_{pqs}^{e,u} = \delta_{p,\psi} z_{pqs}^{\sigma\bar{\sigma},u}, \qquad & p, q, s, u \in L_{\mathrm{TC}}. \end{aligned} \tag{96}$$

The nonzero components of $z^{J_{\mathrm{TC}}}$ tensors are

$$z_{111}^{1,1} = z_{11\psi}^{1,\psi} = 1, \tag{97}$$

$$z_{\psi\psi 1}^{e,\psi} = z_{\psi\psi\psi}^{e,1} = 1, \tag{98}$$

$$z_{111}^{m,1} = -z_{11\psi}^{m,1} = 1, \tag{99}$$

$$z_{\psi\psi 1}^{\epsilon,\psi} = -z_{\psi\psi\psi}^{\epsilon,1} = 1. \tag{100}$$

These $z^{J_{\mathrm{TC}}}$ tensors are the four minimal solutions to the equation

$$\frac{\delta_{j,t} N_{rs}^t}{d_t} z_{pqt}^{J_{\mathrm{TC}};w} = \sum_{ulv \in L_{\mathrm{TC}}} d_u d_v z_{lqr}^{J_{\mathrm{TC}};v} z_{pls}^{J_{\mathrm{TC}};u} G_{pwu}^{rst} G_{qwv}^{srj} G_{rvw}^{sul}, \tag{101}$$

with all indices in $L_{\mathrm{TC}} = \{1, \psi\}$.

Note that although the doubled-Ising tensor $z^{\sigma\bar{\sigma}}$ (91) also solves Eq. (101), it is not a minimal solution but the sum of two minimal solutions $z^e$ and $z^m$:

$$z_{pqs}^{\sigma\bar{\sigma},u} = z_{pqs}^{m,u} + z_{pqs}^{e,u}, \qquad p, q, r, s \in L_{\mathrm{TC}}. \tag{102}$$

### D.3   Nonzero components of $z^{J_{\mathrm{DW}}}$ tensors in the gapped domain wall

The tensors $z^{J_{\mathrm{DW}}}$, $J_{\mathrm{DW}} \in \{1, e, m, \epsilon, \chi, \bar{\chi}\}$, are

$$
z^{1,u}_{pqs} = z^{1\bar{1},u}_{pqs} = z^{\psi\bar{\psi},u}_{pqs}, \qquad z^{\epsilon,u}_{pqs} = z^{\psi\bar{1},u}_{pqs} = z^{1\bar{\psi},u}_{pqs}, \qquad z^{m,u}_{pqs} = \delta_{p,1} z^{\sigma\bar{\sigma},u}_{pqs}, \qquad z^{e,u}_{pqs} = \delta_{p,\psi} z^{\sigma\bar{\sigma},u}_{pqs},
$$

$$
z^{\chi,u}_{pqs} = z^{\sigma\bar{1}1,u}_{pqs} = z^{\sigma\bar{\psi},u}_{pqs}, \qquad z^{\bar{\chi},u}_{pqs} = z^{1\bar{\sigma},u}_{pqs} = z^{\psi\bar{\sigma},u}_{pqs}, \qquad p, q, u \in L_{\mathrm{DI}}, \qquad s \in L_{\mathrm{TC}}, \tag{103}
$$

where the nonzero components are

$$
z^{1,1}_{111} = z^{1,\psi}_{11\psi} = 1, \tag{104}
$$

$$
z^{e,\psi}_{\psi\psi 1} = z^{e,1}_{\psi\psi\psi} = 1, \tag{105}
$$

$$
z^{m,1}_{111} = -z^{m,1}_{11\psi} = 1, \tag{106}
$$

$$
z^{\epsilon,\psi}_{\psi\psi 1} = -z^{\epsilon,1}_{\psi\psi\psi} = 1, \tag{107}
$$

$$
z^{\chi,\sigma}_{\sigma\sigma 1} = 1, \qquad z^{\chi,\sigma}_{\sigma\sigma\psi} = i, \tag{108}
$$

$$
z^{\bar{\chi},\sigma}_{\sigma\sigma 1} = 1, \qquad z^{\bar{\chi},\sigma}_{\sigma\sigma\psi} = -i. \tag{109}
$$

The tensors $z^{J_{\mathrm{DW}}}$ are the 6 minimal solutions to the equation

$$
\frac{\delta_{j,t} N^t_{rs}}{d_t} z^{J_{\mathrm{DW}};w}_{pqt} = \sum_{ulv \in L_{\mathrm{DI}}} d_u d_v z^{J_{\mathrm{DW}};v}_{lqr} z^{J_{\mathrm{DW}};u}_{pls} G^{rst}_{pwu} G^{srj}_{qwv} G^{sul}_{rvw}, \tag{110}
$$

with all indices in $L_{\mathrm{DI}}$ except that $r, s, t \in L_{\mathrm{TC}} = \{1, \psi\}$.

## E   Measuring elementary excitation states by local operators

### E.1   Local operators in the doubled Ising phase

Since our model stems from the extended LW model describing the doubled Ising phase, we first focus on the local operators in the doubled Ising phase. In the doubled Ising phase, the local operators $B^{psqu}_P$ are defined by

The local operators $B_P^{\mathrm{psqu}}$ preserve the anyon species $J_{\mathrm{DI}}$ of the elementary excitation states $|J_{\mathrm{DI}}; p, r\rangle_{\mathrm{DI}}$ but change the charges of the doubled-Ising quasiparticles $(J_{\mathrm{DI}}, p)$ in plaquettes $P$.

$$
B_P^{psqu} \left| \begin{array}{c} \end{array} (J_{\mathrm{DI}}, p') \rule{1cm}{0.5pt} (J_{\mathrm{DI}}, r) \right\rangle \;\propto\; \delta_{p,p'} \left| \begin{array}{c} \end{array} (J_{\mathrm{DI}}, q) \rule{1cm}{0.5pt} (J_{\mathrm{DI}}, r) \right\rangle . \tag{112}
$$

There are in total 12 local operators $B_P^{psqu}$ acting on plaquette $P$:

$$
\begin{aligned}
& B_P^{1111}, && B_P^{1\psi1\psi}, && B_P^{1\sigma1\sigma}, && B_P^{1\sigma\psi\sigma}, && B_P^{\psi1\psi\psi}, && B_P^{\psi\psi\psi1}, \\
& B_P^{\psi\sigma\psi\sigma}, && B_P^{\psi\sigma1\sigma}, && B_P^{\sigma1\sigma\sigma}, && B_P^{\sigma\psi\sigma\sigma}, && B_P^{\sigma\sigma1}, && B_P^{\sigma\sigma\sigma\psi}.
\end{aligned} \tag{113}
$$

## E.2 Measurement operators in the doubled Ising phase

Now we define the measurement operators of the elementary excitation states in the doubled Ising phase via the local operators defined above. Since the doubled-Ising elementary excitation states $|J_{\mathrm{DI}}; p, q\rangle_{\mathrm{DI}}$ are determined by quasiparticles $(J_{\mathrm{DI}}, p)$ and $(J_{\mathrm{DI}}, q)$ therein, to measure the elementary excitation states, we only need to detect the quasiparticles in the plaquettes.

The measurement operators $\Pi_P^{J_{\mathrm{DI}}, p}$ of quasiparticles $(J_{\mathrm{DI}}, p)$ in plaquette $P$ is a linear composition of local operators

$$
\Pi_P^{J_{\mathrm{DI}}, p} := \sum_{su} \pi_{psu}^{J_{\mathrm{DI}}} B_P^{pspu}. \tag{114}
$$

Here the coefficients $\pi_{psu}^{J_{\mathrm{DI}}}$ satisfy

$$
\frac{\pi_{psu}^{J_{\mathrm{DI}}}}{\pi_{p1p}^{J_{\mathrm{DI}}}} = \frac{d_s d_u}{d_p} z_{pps}^{J_{\mathrm{DI}}; u} \tag{115}
$$

where $\pi_{p1p}^J$ is a normalization factor, such that

$$
\Pi_P^{J_{\mathrm{DI}}, p} \left| \begin{array}{c} \end{array} (J'_{\mathrm{DI}}, p') \rule{1cm}{0.5pt} (J_{\mathrm{DI}}, r) \right\rangle = \delta_{p,p'} \delta_{J_{\mathrm{DI}}, J'_{\mathrm{DI}}} \left| \begin{array}{c} \end{array} (J'_{\mathrm{DI}}, p') \rule{1cm}{0.5pt} (J_{\mathrm{DI}}, r) \right\rangle . \tag{116}
$$

The measurement operators $\Pi_P^{J_{\mathrm{DI}}}$ of anyon species $J_{\mathrm{DI}}$ are thus

$$
\Pi_P^{J_{\mathrm{DI}}} = \sum_{p \in J_{\mathrm{DI}}} \Pi_P^{J_{\mathrm{DI}}, p}, \tag{117}
$$

where $p \in J_{\mathrm{DI}}$ are the charges of $J_{\mathrm{DI}}$ anyons, i.e., there exist $q, s, u \in L_{\mathrm{DI}}$, such that $z_{pqs}^{J_{\mathrm{DI}}, u} \neq 0$.

### E.3 Local operators and measurement operators in the toric code domain and the gapped domain wall

The local operators in our model are projected from the local operators (111) in the doubled Ising phase, while the measurement operators of quasiparticles in our model are projected from the doubled-Ising measurement operators (114).

There are four local operators acting on the TC plaquette $P$:

$$P_{\text{eff}} B_P^{1111} P_{\text{eff}}, \qquad P_{\text{eff}} B_P^{1\psi 1\psi} P_{\text{eff}}, \qquad P_{\text{eff}} B_P^{\psi 1\psi\psi} P_{\text{eff}}, \qquad P_{\text{eff}} B_P^{\psi\psi\psi 1} P_{\text{eff}}, \tag{118}$$

which comprise the measurement operators of the four quasiparticles $(J_{\text{TC}}, p)$ in the toric code domain

$$\Pi_P^{1,1} = \frac{1}{2} P_{\text{eff}} \left( B_P^{1111} + B_P^{1\psi 1\psi} \right) P_{\text{eff}},$$

$$\Pi_P^{m,1} = \frac{1}{2} P_{\text{eff}} \left( B_P^{1111} - B_P^{1\psi 1\psi} \right) P_{\text{eff}},$$

$$\Pi_P^{e,\psi} = \frac{1}{2} P_{\text{eff}} \left( B_P^{\psi 1\psi\psi} + B_P^{\psi\psi\psi 1} \right) P_{\text{eff}},$$

$$\Pi_P^{\epsilon,\psi} = \frac{1}{2} P_{\text{eff}} \left( B_P^{\psi 1\psi\psi} - B^{\psi\psi\psi 1} \right)_P P_{\text{eff}}. \tag{119}$$

In the gapped domain wall, since $s$ is restricted to $L_{\text{TC}}$, there are 6 local operators:

$$P_{\text{eff}} B_P^{1111} P_{\text{eff}}, \qquad\qquad P_{\text{eff}} B_P^{1\psi 1\psi} P_{\text{eff}}, \qquad\qquad P_{\text{eff}} B_P^{\psi 1\psi\psi} P_{\text{eff}},$$

$$P_{\text{eff}} B_P^{\psi\psi\psi 1} P_{\text{eff}}, \qquad\qquad P_{\text{eff}} B_P^{\sigma 1\sigma\sigma} P_{\text{eff}}, \qquad\qquad P_{\text{eff}} B_P^{\sigma\psi\sigma\sigma} P_{\text{eff}}. \tag{120}$$

They comprise the measurement operators of the domainwall quasiparticles $(J_{\text{DW}}, p)$.

$$\Pi_P^{1,1} = \frac{1}{2} P_{\text{eff}} \left( B_P^{1111} + B_P^{1\psi 1\psi} \right) P_{\text{eff}},$$

$$\Pi_P^{m,1} = \frac{1}{2} P_{\text{eff}} \left( B_P^{1111} - B_P^{1\psi 1\psi} \right) P_{\text{eff}},$$

$$\Pi_P^{e,\psi} = \frac{1}{2} P_{\text{eff}} \left( B_P^{\psi 1\psi\psi} + B_P^{\psi\psi\psi 1} \right) P_{\text{eff}},$$

$$\Pi_P^{\epsilon,\psi} = \frac{1}{2} P_{\text{eff}} \left( B_P^{\psi 1\psi\psi} - B_P^{\psi\psi\psi 1} \right) P_{\text{eff}},$$

$$\Pi_P^{\chi,\sigma} = \frac{\sqrt{2}}{2} P_{\text{eff}} \left( B_P^{\sigma 1\sigma\sigma} + i B_P^{\sigma\psi\sigma\sigma} \right) P_{\text{eff}},$$

$$\Pi_P^{\bar{\chi},\sigma} = \frac{\sqrt{2}}{2} P_{\text{eff}} \left( B_P^{\sigma 1\sigma\sigma} - i B_P^{\sigma\psi\sigma\sigma} \right) P_{\text{eff}}. \tag{121}$$

Using the measurement operators (114), (119) and (121) in different areas of our model, we can measure the quasiparticle species of our model. See Table 1, 2, 4, 5 and 6.

## F The algebra of the noncontractible loop operators

### F.1 The multiplications of noncontractible loop operators

The loop operators $W_H^{J_{\text{DI}} - J_{\text{TC}}}$ (47) and $W_V^{J_{\text{DW}}}$ (48) generate a 36-dimensional algebra $\mathcal{A}$. Here, we list the multiplications of these loop operators, which completely determine this algebra.

The six $H$-loop operators $W_H^{J_{\mathrm{DI}}-J_{\mathrm{TC}}}$ are commutative:

$$\left(W_H^{1\bar\psi-\epsilon}\right)^2 = \left(W_H^{\psi\bar 1-\epsilon}\right)^2 = W_H^{1\bar 1-1} = I, \qquad W_H^{\psi\bar 1-\epsilon}W_H^{1\bar\psi-\epsilon} = W_H^{\psi\bar\psi-1},$$

$$W_H^{\psi\bar 1-\epsilon}W_H^{\sigma\bar\sigma-e} = W_H^{1\bar\psi-\epsilon}W_H^{\sigma\bar\sigma-e} = W_H^{\sigma\bar\sigma-m}, \qquad \left(W_H^{\sigma\bar\sigma-e}\right)^2 = \frac{W_H^{1\bar 1-1}+W_H^{\psi\bar\psi-1}}{2}. \qquad (122)$$

The six $V$-loop operators $W_V^{J_{\mathrm{DW}}}$ along the gapped domain wall are also commutative:

$$(W_V^e)^2 = (W_V^m)^2 = W_V^1 = I, \qquad W_V^e W_V^m = W_V^\epsilon,$$
$$W_V^e W_V^\chi = W_V^m W_V^\chi = W_V^{\bar\chi}, \qquad (W_V^\chi)^2 = W_V^1 + W_D^\epsilon. \qquad (123)$$

Multiplying $W_V^{J_{\mathrm{DW}}}$ and $W_H^{J_{\mathrm{DI}}-J_{\mathrm{TC}}}$ generate the additional 25 linearly independent symmetry operators.

$$W_H^{\sigma\bar\sigma\,\text{-}\,m}W_V^m = W_V^m W_H^{\sigma\bar\sigma\,\text{-}\,m},$$
$$W_H^{\sigma\bar\sigma\,\text{-}\,e}W_V^e = W_V^e W_H^{\sigma\bar\sigma\,\text{-}\,e},$$
$$W_H^{\sigma\bar\sigma\,\text{-}\,e}W_V^m = -W_V^m W_H^{\sigma\bar\sigma\,\text{-}\,e},$$
$$W_H^{\sigma\bar\sigma\,\text{-}\,m}W_V^e = -W_V^e W_H^{\sigma\bar\sigma\,\text{-}\,m},$$
$$W_H^{\sigma\bar\sigma\,\text{-}\,e}W_V^\epsilon = -W_V^\epsilon W_H^{\sigma\bar\sigma\,\text{-}\,e},$$
$$W_H^{\sigma\bar\sigma\,\text{-}\,m}W_V^\epsilon = -W_V^\epsilon W_H^{\sigma\bar\sigma\,\text{-}\,m},$$
$$W_H^{\psi\bar\psi\,\text{-}\,1}W_V^\chi = -W_V^\chi W_H^{\psi\bar\psi\,\text{-}\,1},$$
$$W_H^{\psi\bar\psi\,\text{-}\,1}W_V^{\bar\chi} = -W_V^{\bar\chi} W_H^{\psi\bar\psi\,\text{-}\,1},$$
$$W_H^{\psi\bar\psi\,\text{-}\,1}W_V^m = W_V^m W_H^{\psi\bar\psi\,\text{-}\,1},$$
$$W_H^{\psi\bar 1\,\text{-}\,\epsilon}W_V^\epsilon = W_V^\epsilon W_H^{\psi\bar 1\,\text{-}\,\epsilon},$$
$$W_H^{\psi\bar 1\,\text{-}\,\epsilon}W_V^\chi = -W_V^\chi W_H^{\psi\bar 1\,\text{-}\,\epsilon},$$
$$W_H^{\psi\bar 1\,\text{-}\,\epsilon}W_V^{\bar\chi} = -W_V^{\bar\chi} W_H^{\psi\bar 1\,\text{-}\,\epsilon},$$
$$W_H^{1\bar\psi\,\text{-}\,\epsilon}W_V^\chi = -W_V^\chi W_H^{1\bar\psi\,\text{-}\,\epsilon},$$
$$W_H^{1\bar\psi\,\text{-}\,\epsilon}W_V^{\bar\chi} = -W_V^{\bar\chi} W_H^{1\bar\psi\,\text{-}\,\epsilon},$$
$$W_H^{\psi\bar 1\,\text{-}\,\epsilon}W_V^m = -W_V^m W_H^{\psi\bar 1\,\text{-}\,\epsilon},$$
$$W_H^{1\bar\psi\,\text{-}\,\epsilon}W_V^m = -W_V^m W_H^{1\bar\psi\,\text{-}\,\epsilon},$$
$$W_H^{\psi\bar 1\,\text{-}\,\epsilon}W_V^e = -W_V^e W_H^{\psi\bar 1\,\text{-}\,\epsilon},$$
$$W_H^{\sigma\bar\sigma\,\text{-}\,m}W_V^\chi,$$
$$W_V^\chi W_H^{\sigma\bar\sigma\,\text{-}\,m},$$
$$W_H^{\sigma\bar\sigma\,\text{-}\,e}W_V^\chi,$$
$$W_V^\chi W_H^{\sigma\bar\sigma\,\text{-}\,e},$$
$$W_H^{\sigma\bar\sigma\,\text{-}\,m}W_V^{\bar\chi},$$
$$W_V^{\bar\chi} W_H^{\sigma\bar\sigma\,\text{-}\,m},$$

$$W_H^{\sigma\bar{\sigma}-e} W_V^{\bar{\chi}},$$

$$W_D^{\bar{\chi}} W_H^{\sigma\bar{\sigma}-e}. \tag{124}$$

All other multiplications of operators are not linearly independent:

$$W_H^{\psi\bar{\psi}-1} W_V^e = W_V^e W_H^{\psi\bar{\psi}-1} = W_H^{\psi\bar{\psi}-1} W_V^m + W_V^e - W_V^m,$$

$$W_H^{1\bar{\psi}-\epsilon} W_V^\epsilon = W_V^\epsilon W_H^{1\bar{\psi}-\epsilon} = W_H^{\psi\bar{1}-\epsilon} W_V^\epsilon + W_H^{1\bar{\psi}-\epsilon} - W_H^{\psi\bar{1}-\epsilon},$$

$$W_H^{1\bar{\psi}-\epsilon} W_V^e = -W_V^e W_H^{1\bar{\psi}-\epsilon} = W_H^{\psi\bar{1}-\epsilon} W_V^e + W_H^{1\bar{\psi}-\epsilon} W_V^e - W_H^{\psi\bar{1}-\epsilon} W_V^m,$$

$$W_V^\chi W_H^{\sigma\bar{\sigma}-e} W_V^\chi = W_V^\chi W_H^{\sigma\bar{\sigma}-m} W_V^\chi = 0. \tag{125}$$

## F.2 Generating the entire ground-state subspace

Finally, we prove Eq. (49): For any given two ground states $|\Phi\rangle \in \mathcal{H}_{\text{eff}}$ and $|\Phi'\rangle \in \mathcal{H}_{\text{eff}}$ of our model on the torus, there exists an operator $W \in \mathcal{A}$, such that

$$|\Phi'\rangle = W |\Phi\rangle. \tag{126}$$

Since the ribbon operators in our model are projected from the doubled-Ising ribbon operators in the doubled Ising phase, the doubled-Ising loop operators in our model on the torus — the special cases of ribbon operators — are also projections of the doubled-Ising loop operators along the same paths:

$$W_V^1 = P_{\text{eff}} W_V^{1\bar{1}} P_{\text{eff}} = P_{\text{eff}} W_V^{\psi\bar{\psi}} P_{\text{eff}},$$

$$W_V^\epsilon = P_{\text{eff}} W_V^{\psi\bar{1}} P_{\text{eff}} = P_{\text{eff}} W_V^{1\bar{\psi}} P_{\text{eff}},$$

$$W_V^m + W_V^e = P_{\text{eff}} W_V^{\sigma\bar{\sigma}} P_{\text{eff}},$$

$$W_V^\chi = P_{\text{eff}} W_V^{\sigma\bar{1}} P_{\text{eff}} = P_{\text{eff}} W_V^{\sigma\bar{\psi}} P_{\text{eff}},$$

$$W_V^{\bar{\chi}} = P_{\text{eff}} W_V^{1\bar{\sigma}} P_{\text{eff}} = P_{\text{eff}} W_V^{\psi\bar{\sigma}} P_{\text{eff}},$$

$$W_H^{1\bar{1}-1} = P_{\text{eff}} W_H^{1\bar{1}} P_{\text{eff}},$$

$$W_H^{\psi\bar{\psi}} = P_{\text{eff}} W_H^{\psi\bar{\psi}} P_{\text{eff}},$$

$$W_H^{\sigma\bar{\sigma}-m} + W_H^{\sigma\bar{\sigma}-e} = P_{\text{eff}} W_H^{\sigma\bar{\sigma}} P_{\text{eff}},$$

$$W_H^{\psi\bar{1}-\epsilon} = P_{\text{eff}} W_H^{\psi\bar{1}} P_{\text{eff}},$$

$$W_H^{1\bar{\psi}-\epsilon} = P_{\text{eff}} W_H^{1\bar{\psi}} P_{\text{eff}}. \tag{127}$$

Therefore, the algebra $\mathcal{A}$ in our model satisfies

$$\mathcal{A} = P_{\text{eff}} \mathcal{A}^{\text{DI}} P_{\text{eff}}, \tag{128}$$

where the algebra $\mathcal{A}^{\text{DI}}$ is generated by all noncontractible loop operators $W_D^{J_{\text{DI}}}, W_H^{J_{\text{DI}}}$ in the doubled Ising phase on the torus along $H$-loop and $V$-loop.

On the other hand, in the doubled Ising phase, the projector

$$P_0^{\text{DI}} = \prod_P B_P^{\text{DI}} \prod_V Q_V \tag{129}$$

projects the total Hilbert space $\mathcal{H}$ to the doubled-Ising ground-state subspace $\mathcal{H}_0^{\text{DI}}$:

$$P_0^{\text{DI}} \mathcal{H} = \mathcal{H}_0^{\text{DI}}. \tag{130}$$

In our model, the projector

$$P_0 = \prod_{P \in \mathrm{DI}} B_P^{\mathrm{DI}} \prod_{P \in \mathrm{DW}} B_P^{\mathrm{DW}} \prod_{P \in \mathrm{TC}} B_P^{\mathrm{TC}} \prod_V Q_V \tag{131}$$

projectes the effecitve Hilbert space $\mathcal{H}_{\mathrm{eff}}$ to the ground-state subspace $\mathcal{H}_0$. Note that $P_0 = P_{\mathrm{eff}} P_0^{\mathrm{DI}} P_{\mathrm{eff}}$, the projector $P_{\mathrm{eff}}$ projects $\mathcal{H}_0^{\mathrm{DI}}$ to $\mathcal{H}_0$:

$$P_{\mathrm{eff}} \mathcal{H}_0^{\mathrm{DI}} = P_{\mathrm{eff}} \mathcal{H}_0^{\mathrm{DI}}. \tag{132}$$

Therefore, for any two ground states $|\Phi\rangle$ and $|\Phi'\rangle$ of our model, there exist doubled-Ising ground states $|\Phi\rangle_{\mathrm{DI}}$ and $|\Phi'\rangle_{\mathrm{DI}}$, such that

$$|\Phi\rangle = |\Phi\rangle_{\mathrm{DI}}, \qquad |\Phi'\rangle = |\Phi'\rangle_{\mathrm{DI}}. \tag{133}$$

Since $\mathcal{H}_0^{\mathrm{DI}}$ is generated by the algebra $\mathcal{A}^{\mathrm{DI}}$ given any doubled-Ising ground state $|\Phi\rangle_{\mathrm{DI}}$, there exists a doubled-Ising operator $W^{\mathrm{DI}} \in \mathcal{A}^{\mathrm{DI}}$, such that

$$|\Phi'\rangle_{\mathrm{DI}} = W^{\mathrm{DI}} |\Phi\rangle_{\mathrm{DI}}. \tag{134}$$

As their projections,

$$|\Phi'\rangle = \left( P_{\mathrm{eff}} W^{\mathrm{DI}} P_{\mathrm{eff}} \right) |\Phi\rangle_{\mathrm{DI}}, \tag{135}$$

where $P_{\mathrm{eff}} W^{\mathrm{DI}} P_{\mathrm{eff}} \in \mathcal{A}$. Therefore, the algebra $\mathcal{A}$ generates the entire ground-state subspace $\mathcal{H}_0$ of our model given any ground state $|\Phi\rangle$.

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
