# Peer review of "Exactly solvable Hamiltonian model of the doubled Ising and $\mathbb{Z}_2$ toric code topological phases separated by a gapped domain wall via anyon condensation"

_SciPost Physics_

## Round 1 · Referee Report · Anonymous · 2023-5-15

Strengths
1. Clearly written.
2. Well-formulated problem.
3. Provides insight into interfaces of topological ordered phases, the analysis of which has mostly been on interfaces of topological and trivial phases.
Weaknesses
1. Unreadable for a non-expert. Knowledge of one of the authors previous works on interfaces of topological phases is mandatory to follow this work.
2. Not properly motivated. Why choose an interface of doubled Ising and toric code phases?
Report
The authors address an interesting problem of an interface of topological phases and quantify the anyonic content, S and T matrices and ground state degeneracies in the presence of the interface.
While the paper is clearly written and the results are clearly stated, it is not clear what the main motivation of the work is.
1) Why choose the doubled Ising and the toric code phases to construct interface?
2) Is there an underlying theory of what kind of interfaces arise between the different phases described by different Levin-Wen Hamiltonians?
3) Is it important that one of the phases is non-abelian?
The proposed work is very far from anything related to experiments. Without proper motivation, it is very hard to comprehend why the authors chose the current problem.
The current work builds on earlier works on interfaces between topological phases. It is an interesting work, but it is not clear why it should be published in Scipost Physics since (as far as I can tell), it does not meet any of the 'Expectations' criteria. As such, I do not recommend publication in Scipost Physics.
However, if the authors can address my critique satisfactorily, I will be happy to reconsider my decision.
Thank the referee for taking the time to review our manuscript and for providing valuable feedback. We are grateful for the insightful comments and the opportunity to address them.
Based on the feedback from the referee's report, we will refine our manuscript, especially in the abstract and introduction sections, with the aim of more explicitly stating our motivation. We will also revise the title to "Characteristic Properties of a Composite System of Topological Phases Separated by Gapped Domain Walls via an Exactly Solvable Hamiltonian Model". We summarize our motivation as follows.
Our motivation is to explore the properties of a generalized topological system consisting of multiple topological orders within subsystems separated by topological interfaces. Previous research has focused on single topologically ordered systems, characterized by their topological properties such as ground-state degeneracy, and the modular S and T matrices. Nevertheless, the characteristic properties of composite systems remain largely uncharted. To unravel such systems, it is profound to see how the properties of a single topological ordered system would adapt in such composite systems, as well as to identify the excitation spectra and quasiparticles, especially those quasiparticles in the gapped domain wall. To facilitate this understanding, we construct an exactly solvable lattice model. For clarity and explicitness, we focus on a special example in our paper. Our model enables us to examine the properties of the composite system through tangible wavefunctions. Further, recognizing that a temporal phase transition triggered by a particular anyon condensation always corresponds to a spatial composite system, we aim to solidify this correlation using our model.
Below, we address each of the referee’s concerns.
(1) Why choose the doubled Ising and the toric code phases to construct the interface?
We appreciate your query regarding our choice of the topological interface between the doubled-Ising and toric code phases. For clarity and explicitness, we chose in our paper to focus on the model in the special case --- the doubled Ising and toric code topological phases with a topological interface in between --- over a more general construction because this example is simple and familiar within the research community. Moreover, the anyon condensation between these two phases encapsulates all the key phenomena --- identification, splitting, and confinement --- of the general theory of anyon condensation, making it an ideal choice for our study. Interestingly, despite the familiarity of this system, our model continues to reveal novel phenomena, such as the detailed mechanics of splitting.
While our study focuses on specific topological phases, we wish to emphasize the universality of our construction methodology and conclusions. Our construction method can be directly utilized for any composite system including a parent topological phase and a child phase separated by topological interfaces, enabling researchers to adapt it based on their interests. Our conclusions describe the characteristic properties of any composite system. For instance, we have obtained the domain-wall S matrix in general composite systems (Arxiv:2305.03766), applicable even to chiral topological systems without a lattice model description.
(2) Is there an underlying theory of what kind of interfaces arise between the different phases described by different Levin-Wen Hamiltonians?
The nature of topological interfaces between different topological phases can be complicated, and a unified mathematical structure is yet not fully established. An algebraic approach to describing the types of topological interfaces that might exist between topological phases has been proposed in [PRL 114.7 (2015): 076402].
Our model, based on the Levin-Wen model, primarily focuses on the topological interfaces between topological phases that can be represented by a lattice model, namely the doubled topological phases.
(3) Is it important that one of the phases is non-abelian?
No. Our methodology is universally applicable to any composite system composed of two topological phases separated by topological interfaces. The only requirement is that one of the topological phases (the child phase) is obtained via a phase transition triggered by a certain anyon condensation in the other parent topological phase.
(4) Unreadable for a non-expert.
We recognize the value of making our research accessible to a broader audience. While our study builds on previous work, particularly Professor Hu's research (PRB 97.19 (2018): 195154) on the extended Levin-Wen model, we have aimed to make our manuscript self-sufficient and easy to comprehend. We will revise our manuscript with the hope that our detailed model definition enables readers to grasp the concepts within this paper alone, without needing prior knowledge of the referenced work.
(5) Far from experiments.
While our work is theoretical, it has potential implications for experimental realizations of topological quantum computation. Recent advancements in topological quantum computation have successfully created non-Abelian topological order on a trapped-ion processor (ArXiv:2305.03766), realized a topologically protected Hadamard gate via braiding Fibonacci anyons on topological boundaries (ArXiv:2210.12145), and suppressed quantum errors by scaling a surface code logical qubit (Nature 614, no. 7949 (2023): 676-681). Our study may extend these efforts by investigating topological interfaces, which are generalizations of topological boundaries that have applications in experimental realizations of defect-based topological quantum computation (Nature Physics 14.2 (2018): 160-165, ArXiv: 1609.02037, Physical Review Letters 119.17 (2017): 170504, ArXiv:1710.07197). We believe that our work will contribute to the ongoing advancements in topological quantum computation.
(6) Does not meet 'Expectations' criteria.
We have carefully reviewed the 'Expectations' criteria for SciPost Physics. We are confident that our work aligns with these guidelines. Our research presents original contributions in composite systems consisting of multiple topological phases, and our findings have potential applications in various domains, both theoretically and experimentally, such as topological quantum computation, topological field theories, and conformal field theories.
In fact, our research has already inspired further novel results, as demonstrated in [Arxiv:2305.03766]. We have defined the domain-wall $S$ matrix in general composite systems and derived the fusion rules of domain-wall quasiparticles and interdomain excitations using formulae analogous to the Verlinde formula. These fusion rules reveal novel and intriguing properties, further highlighting the innovative aspects of topological composite systems.
We hope that these revisions address the referee's concerns and make our manuscript suitable for publication in SciPost Physics. We look forward to further feedback on our revised manuscript.
Best regards,
Yidun Wan, on behalf of all authors
Anonymous on 2023-06-27 [id 3767]
Dear Editors and Referees,
We're pleased to let you know that we have made the revised version of our manuscript available on ArXiv under the reference number 2209.12750. Importantly, the title of our paper has been changed to "Characteristic Properties of a Composite System of Topological Phases Separated by Gapped Domain Walls via an Exactly Solvable Hamiltonian Model".
In order to resubmit this updated version of our manuscript on SciPost Physics, we kindly ask for a recommendation from the Editor-in-charge.
Thank you for your continuous support and consideration.
Best regards,
Yidun Wan, on behalf of all authors

---

## Editorial Decision

unknown